# Tracking multiple conformations occurring on angstrom-and-millisecond scales in single amino-acid-transporter molecules

Yufeng Zhou, John H Lewis, Zhe Lu*

Department of Physiology, Perelman School of Medicine, University of Pennsylvania, Philadelphia, United States

**Abstract** Most membrane protein molecules undergo conformational changes as they transition from one functional state to another one. An understanding of the mechanism underlying these changes requires the ability to resolve individual conformational states, whose changes often occur on millisecond and angstrom scales. Tracking such changes and acquiring a sufficiently large amount of data remain challenging. Here, we use the amino-acid transporter AdiC as an example to demonstrate the application of a high-resolution fluorescence-polarization-microscopy method in tracking multistate conformational changes of a membrane protein. We have successfully resolved four conformations of AdiC by monitoring the emission-polarization changes of a fluorophore label and quantified their probabilities in the presence of a series of concentrations of its substrate arginine. The acquired data are sufficient for determining all equilibrium constants that fully establish the energetic relations among the four states. The $K_D$ values determined for arginine in four individual conformations are statistically comparable to the previously reported overall $K_D$ determined using isothermal titration calorimetry. This demonstrated strong resolving power of the present polarization-microscopy method will enable an acquisition of the quantitative information required for understanding the expected complex conformational mechanism underlying the transporter's function, as well as those of other membrane proteins.

*For correspondence: zhelu@pennmedicine.upenn.edu

**Competing interest:** The authors declare that no competing interests exist.

## Editor's evaluation

This study uses a single-molecule polarization microscopy approach to identify the different conformation states that the arginine/agmatine antiporter AdiC transitions through during transport. Four states are identified and proposed to correspond to the key steps in the transport cycle, including inward-open, inward occluded, outward occluded and outward open, setting the stage for measurements of equilibrium constants and kinetics associated with transport. This is a cutting-edge and challenging approach that offers the potential for obtaining direct information of protein conformational equilibria that will be of interest to anyone studying membrane transport mechanisms.

## Introduction

Biological membranes enclose individual cells and thereby separate them from their environments. Proteins embedded in the membrane play important roles such as the receptor-mediated signal transduction, and the ion-channel- or transporter-mediated movement of inorganic ions or organic molecules across the membrane. To accomplish these tasks, the protein molecules undergo many necessary conformational changes. An understanding of the mechanism underlying these changes requires the ability to resolve individual conformational states. However, three-dimensional (3D) protein-conformational changes are not only rapid but also occur usually on an angstrom scale. For

these reasons, at the single-molecule level, the tasks to reliably resolve the multistate conformational changes of typical proteins, which occur in four dimensions (4D) on angstrom-and-millisecond scales, remain extremely challenging, not to mention the need for acquiring a sufficiently large amount of data for examining complex conformational mechanisms. For example, protein conformational changes have been probed using a fluorescence resonance energy transfer (FRET)-based method, which allows the deduction of distance between a fluorescence donor and an acceptor, generally on a nanometer scale (*Stryer, 1978*). This deduction requires knowing the actual fluorescence transfer efficiency, which is often difficult to assess because it depends on the local environment and, in most cases, on the relative orientation of the fluorophore pair. When a protein exists in multiple states, resolution and unambiguous identification of conformational states are even more challenging.

Importantly, while a protein molecule undergoes conformational changes, some secondary structures, e.g., α-helices, which are spatially constrained by other secondary structures, inevitably adopt unique spatial orientations in each conformational state. Thus, this feature offers the opportunity to track these conformational states, without the need of determining their detailed 3D features, by monitoring such an α-helix's spatial orientation defined in terms of the inclination and rotation angles ($\theta$ and $\varphi$; *Figure 1A*) with a method of adequate resolution.

One effective way to track the orientation change of a protein is to monitor the emission polarization change of a bifunctional rhodamine attached to an α-helix (*Figure 1B*) using a polarization microscope (*Sase et al., 1997*; *Warshaw et al., 1998*; *Ha et al., 1998*; *Adachi et al., 2000*; *Sosa et al., 2001*; *Forkey et al., 2003*; *Beausang et al., 2008*; *Rosenberg et al., 2005*; *Forkey et al., 2005*; *Fourkas, 2001*; *Ohmachi et al., 2012*; *Lippert et al., 2017*; *Lewis and Lu, 2019c*). The polarization of individual emitted photons, unlike their travel direction, is not meaningfully affected by the diffraction caused by a so-called polarization-preserving objective. The documented resolutions of such polarization-based detection of rotation motion within a protein had been ≥25°, estimated on the basis of 2.5 times of the standard deviation (σ) of angle measurements (*Rosenberg et al., 2005*; *Forkey et al., 2005*; *Ohmachi et al., 2012*; *Lippert et al., 2017*). Thus, this technique has been limited to the investigation of proteins that undergo large-angle changes, e.g., the lever-arm of a myosin that rotates as much as 80°. Recently, our group assembled a polarization microscope with four polarized-emission-recording channels and tracked the orientation change of an isolated, soluble domain of the MthK K+ channel via a bifunctional fluorophore label attached to an α-helix within the protein (*Lewis and Lu, 2019c*; *Lewis and Lu, 2019a*; *Lewis and Lu, 2019b*). By finding optimal hardware, devising necessary numerical corrections for certain system parameters, and developing essential analyses, we have achieved an effective σ as low as 2°, translating to 5° resolution for detecting changes in both $\theta$ and $\varphi$. For reference, the estimated median radius of proteins is ~20 Å (*Brocchieri and Karlin, 2005*; *Erickson, 2009*), and a rotation of 5° or 10° of a site 20 Å away from the origin would lead to an 1.7 or 3.5 Å change in the chord distance. Thus, the capability to resolve this small angle change allows one to track protein-conformational changes that occur on an angstrom scale in proteins of typical sizes.

While we have succeeded in tracking the conformational changes in the artificially excised, soluble gating-domain protein of MthK, our main goal here is to further develop this method for tracking the conformational changes in a membrane protein. In principle, we can use a protein artificially engineered for this purpose, or a biological protein minimally or heavily modified to enable the required measurements. Simply put, what is required here is merely a membrane protein that adopts multiple conformational states, which can be reported by a fluorophore label, so that we can determine whether our method can (i) resolve its conformational changes by tracking orientation of the dipole of an attached fluorophore at the expected resolution set by the signal-to-noise ratio (SNR), (ii) collect a required large amount of data, (iii) analyze the data acquired from individual protein molecules that are attached to the support glass with some flexibility, and (iv) demonstrate how to extract all the equilibrium constants that fully determine the energetic relations among the resolved states, relations that constrain an equilibrium model.

We chose the bacterial transporter AdiC protein for its expected adaptation of multiple states that we can use as a convenient preparation to test the resolving capability of our method in a membrane protein (*Gong et al., 2003*; *Iyer et al., 2003*). An understanding of the detailed biological mechanism of AdiC is not the primary goal here.

For a general context, AdiC, a member of the amino-acid and polyamine organocation (APC) transporter superfamily (*Jack et al., 2000*; *Casagrande et al., 2008*; *Bosshart and Fotiadis, 2019*),

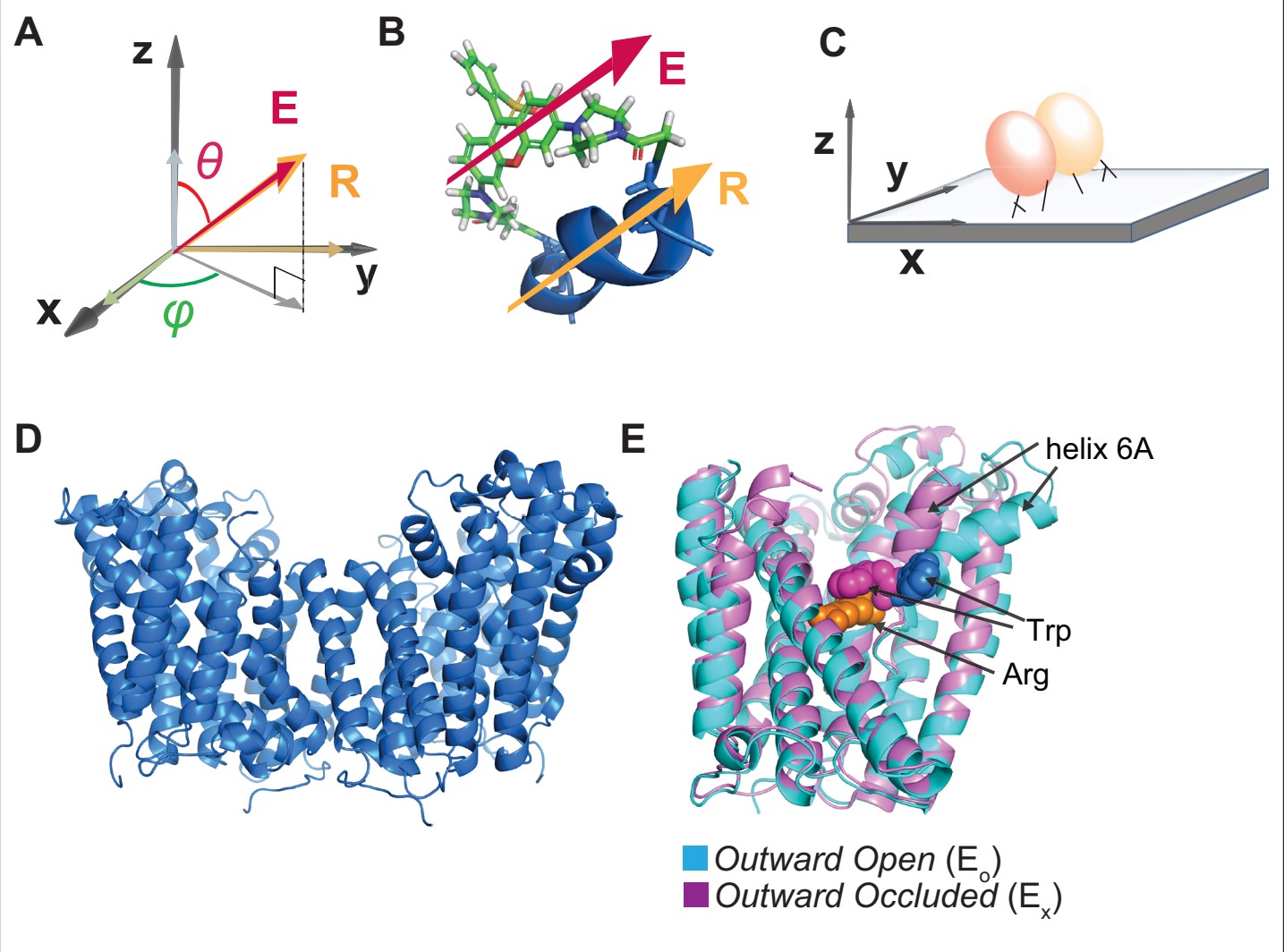

**Figure 1.** Illustration of the attachment of fluorophore to the AdiC protein and the protein to a glass surface. (**A, B**) The orientation of the fluorophore dipole, defined in in terms of $\theta$ and $\varphi$, can be directly related to that of the alpha helix (**A**), to which bifunctional rhodamine is attached via two mutant cysteine residues (**B**) where both orientations of the fluorophore dipole and the helix are indicated by the respective arrows. (**C**) Cartoon illustrating the attachment of an AdiC molecule to a streptavidin-coated coverslip glass via a biotin moiety covalently linked to the N-terminus and two streptavidin-binding tags fused to the N- and C-termini in each of its two subunits, totaling six available sites for binding to streptavidin molecules. (**D**) Structure of AdiC shown as a dimer (PDB: 7O82). (**E**) Spatially aligned structures of $E_O$ and $E_X$ states of AdiC shown with a single subunit (PDB: 3OB6, 3L1L). Helix 6, chosen as a labeling site, is indicated in either structure. The substrate Arg$^+$ (orange) and a Trp residue external to it in the two states (blue and purple) are represented using space-filling models.

The online version of this article includes the following source data and figure supplement(s) for figure 1:

**Figure supplement 1.** Gel filtration chromatography and ITC profiles.

**Figure supplement 1—source data 1.** Data for gel filtration chromatography and ITC profiles.

is a critical component of a proton-extruding system in pathogenic enterobacteria, e.g., *Escherichia*, *Salmonella*, and *Shigella*, which helps the bacteria to survive the insult inflicted by a host's highly acidic gastric defense barrier with a pH value of as low as 2 (*Gong et al., 2003*; *Foster, 2004*; *Fang et al., 2007*; *Iyer et al., 2003*; *Krammer and Prévost, 2019*). AdiC facilitates the movement of arginine (Arg$^+$) into and agmatine (Agm$^{2+}$) out of bacteria, along their gradients. Inside bacteria, Arg$^+$ is rapidly decarboxylated to Agm$^{2+}$ by the enzyme AdiA, consuming a proton (*Gong et al., 2003*; *Foster, 2004*; *Fang et al., 2007*; *Iyer et al., 2003*; *Tsai and Miller, 2013*). An exchange between extracellular Arg$^+$ of a single positive charge and intracellular Agm$^{2+}$ of two charges effectively extrudes H$^+$. The direction of the net exchange of the two substrates is dictated by their natural energy gradient. However,

as an intrinsic property, AdiC can facilitate the movement of a given substrate in either direction, where the transport of one type of substrate does not markedly depend on which type of substrate is on the opposite side.

## Results

### Sample preparations

To monitor the conformational changes of AdiC, we chose to attach a fluorophore label to the surface-exposing helix 6a in one of its two subunits (*Figure 1*; *Gao et al., 2009*; *Fang et al., 2009*; *Gao et al., 2010*; *Kowalczyk et al., 2011*; *Ilgü et al., 2016*, *Ilgü et al., 2021*). Hereafter, unless specified otherwise, AdiC simply refers to one of its two functionally independent subunits (*Figure 1D and E*). Helix 6A is spatially constrained by other secondary structures, and moves along with them, adopting differing spatial orientations between two different known structural states of AdiC (*Figure 1E*). A bifunctional rhodamine molecule was attached via two mutant cysteine residues, spaced seven residues apart, to helix 6a in the region extracellular to the substrate-binding site to avoid affecting the binding affinity (see 'Discussion'). Such an attachment aligned the fluorophore dipole along the axis of the helix (*Corrie et al., 1998*; *Figure 1B*). Under the same labeling condition, there was little detectable fluorescent labeling in the absence of the mutant cysteine residues (*Figure 1—figure supplement 1A and B*). To minimize the background labeling, we removed two native cysteine residues. Removal of native cysteine residues in AdiC has been shown to have very limited impacts on its function (*Tsai et al., 2012*).

For microscopic examination, individual AdiC protein molecules were inserted into nanodiscs (*Ritchie et al., 2009*; *Denisov et al., 2019*). For attaching the protein molecules to streptavidin adhered to the polylysine-coated surface of a piece of coverslip glass, the protein was made to contain a biotin-moiety covalently linked to the N-terminus and the streptavidin-binding tags linked to the N- and C-termini in each of its two functionally independent and structurally symmetric subunits, totaling six sites available for the binding of streptavidin molecules (*Figure 1C* and *Figure 1—figure supplement 1C*; 'Materials and methods'). Assessed with isothermal titration calorimetry (ITC), the protein resulting from the cDNA construct genetically engineered for the present purpose exhibited a $K_D$ of 104 µM for Arg$^+$ (*Figure 1—figure supplement 1D*), which is within the previously reported range of 32–204 µM for AdiC (*Fang et al., 2007*; *Casagrande et al., 2008*; *Gao et al., 2010*; *Tsai et al., 2012*; *Wang et al., 2014*). To minimize the potential constraint of the tag-mediated attachment on the conformational movement of the protein, we included multiple flexible spacer sequences between (i) the Avi and Strep II tags, (ii) these tags and the N-terminus of AdiC, (iii) the thrombin cleavage and Strep II sequences, and (iv) these two sequences and the C-terminus (*Figure 1—figure supplement 1C*). Consequently, the inclination angle $\theta$ varied considerably among molecules. Thus, we needed to develop the analytical method discussed below to spatially align individual molecules.

The density of protein molecules on a cover slip was sufficiently low such that individual fluorescent particles could be readily resolved spatially on microscopic images. The probability of individual AdiC molecules being attached with a single fluorophore was optimized with an empirical protein-to-fluorophore ratio during the labeling procedure, and such molecules were identified during the offline analysis on the basis of a single-step bleaching of fluorescence.

Reconstitution of the protein into lipid-containing nanodiscs, tags, attachment, mutations, and the fluorophore label are all necessary components of the present method, potentially contributing to system errors. Currently, there are no other techniques that can be used to assess, under the same conditions, the ultimate impact of these factors on AdiC, which highlights the need to develop the present method. In terms of energetic impact, a practically relevant question would be whether the $K_D$ values estimated using the present method with all those potentially impactful factors are comparable to those values estimated for the wild-type protein by means of an already established method, such as ITC (see 'Discussion'). As such, we would only know in the end whether the estimates by our method were valid, underscoring the importance of a judicious deliberation in designing the preparation and experiments on the basis of available information, which will help to maximize the chance to obtain a successful outcome.

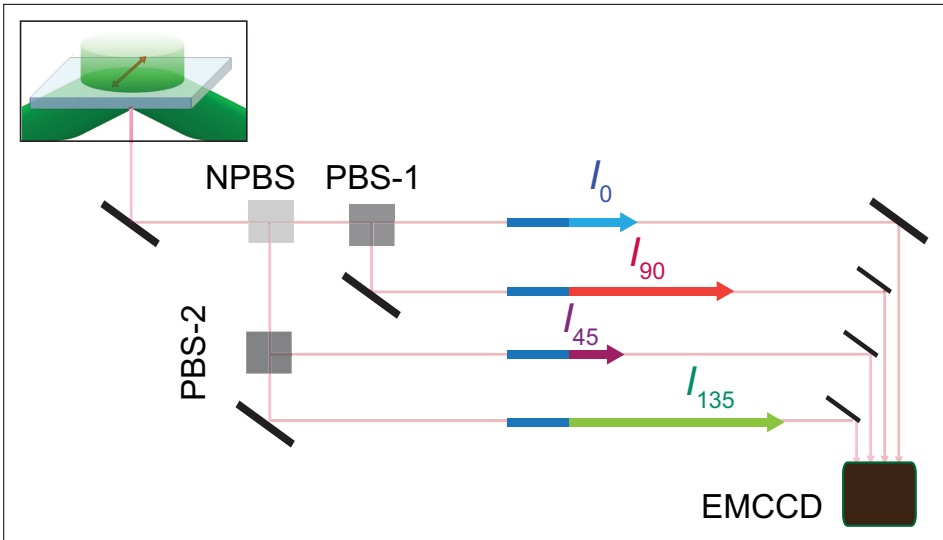

**Figure 2.** Schematic for four polarized emission intensities collected via a microscope and imaged on an EMCCD camera. Photons, emitted from a fluorophore excited by a circularly polarized laser beam, are collected by an objective and directed to a non-polarizing beam splitter (NPBS) that splits it evenly to two beams. Beam 1 is further split into 0 and 90° polarized components ($I_0$ and $I_{90}$) with a glass (N–SF1) polarizing beam splitter (PBS-1), and beam 2 into 45 and 135° components ($I_{45}$ and $I_{135}$) using a wire grid polarizing beam splitter (PBS-2). These four beams are aligned along one path using pick-off mirrors and directed onto separate sections of an EMCCD camera.

## Fluorescence intensity recordings

The fluorophores attached to individual AdiC protein molecules were excited by an evenescent field generated by a circularly polarized laser beam under a total internal reflection (TIR) condition. The light emitted from individual attached fluorophores was captured via the objective of a TIR fluorescence (TIRF) microscope. To assess the polarization of the captured fluorescence light, we first split it into two equal portions and then further split one portion into 0° and 90° polarized components ($I_0$ and $I_{90}$) and the other portion into 45° and 135° components ($I_{45}$ and $I_{135}$) (*Figures 2 and 3*; *Lewis and Lu, 2019c*). These polarized components of fluorescence were recorded with an electron-multiplying charge-coupled device (EMCCD) camera. Effectively, the intensity counts recorded in the four polarization channels encode the full 3D information regarding the orientation of the fluorophore dipole.

We integrated the individual intensity images (*Figure 3A*) and plotted the resulting values against time. As an example, a set of intensity traces for a molecule, each examined in the absence or the presence of 0.75 mM Arg⁺, is shown in *Figure 3B and C*. From these traces, we calculated the trace of total emitted intensity ($I_{tot}$) using *Equation 12* (equations with a number greater than 6 are given in 'Materials and methods'; the bleaching step is shown in *Figure 3—figure supplement 1*).

From *Equations 1 and 2* derived for ideal conditions, one can see that the orientation of the tracked fluorophore would be specifically reflected by the relative intensities, or underlying photon counts, of its four polarized components (*Lewis and Lu, 2019c*; for the solution of three channels, see *Fourkas, 2001*).

$$\varphi = \tfrac{1}{2} \tan^{-1} \frac{I_{45} - I_{135}}{I_0 - I_{90}} \tag{1}$$

$$\theta = \sin^{-1}\left[ 2\sqrt{\frac{\sqrt{(I_0 - I_{90})^2 + (I_{45} - I_{135})^2}}{(I_0 + I_{90} + I_{45} + I_{135})}} \right] \tag{2}$$

Given that either angle would be a function of an intensity ratio, its resolution should be primarily limited by the SNR of intensities. Thus, one can visually notice intramolecular motions that occur on the angstrom scale from the relative variations in the four intensities (*Figure 3A*, *Figure 3—video 1*).

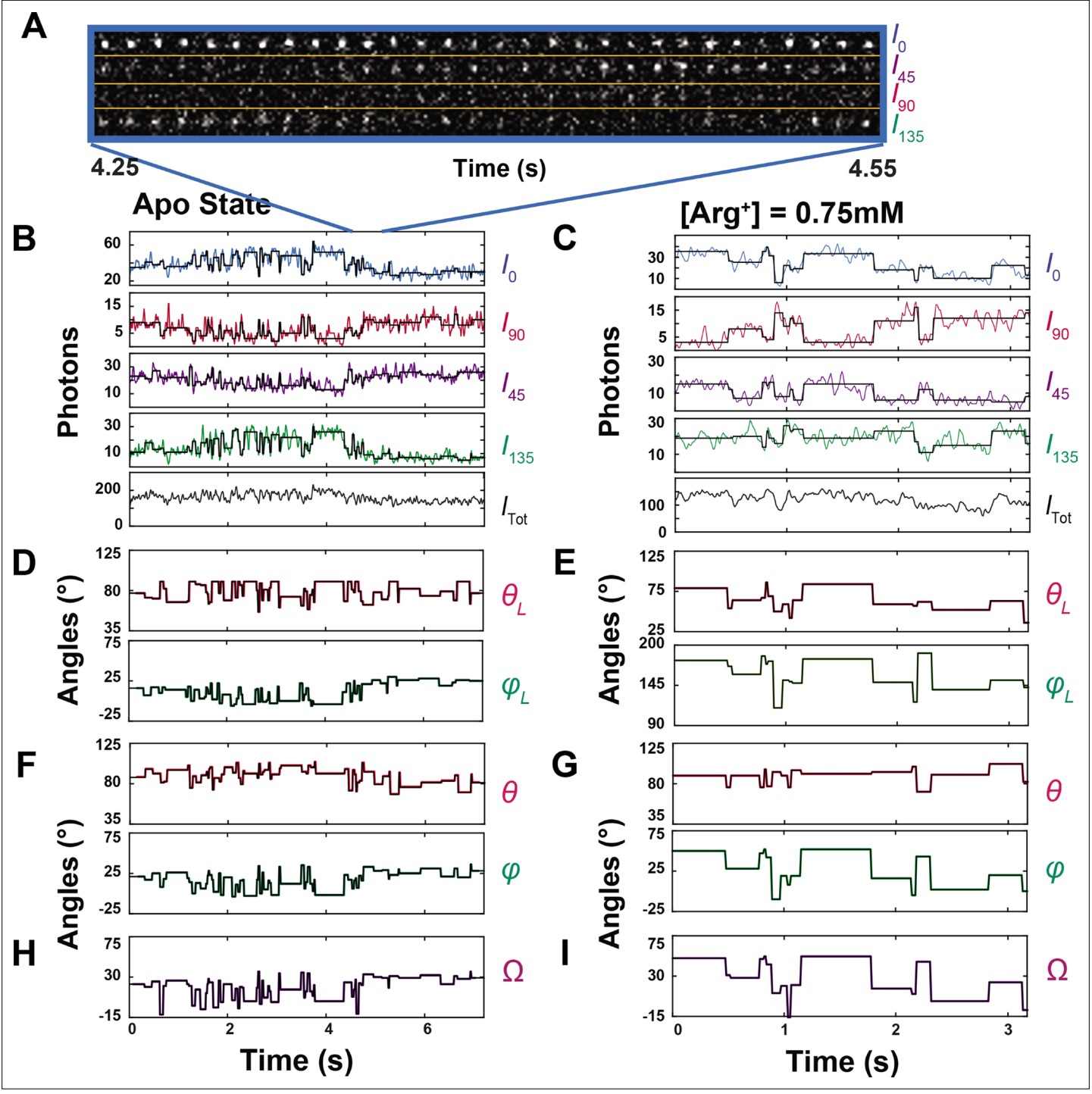

**Figure 3.** Polarized intensity components of single fluorescent particles and $\theta$ and $\varphi$ angles calculated from the components. (**A**) Consecutive frames of four intensity components ($I_0$, $I_{45}$, $I_{90}$ and $I_{135}$) of a bifunctional-rhodamine-labeled apo AdiC molecule captured over a 300 ms interval in a 7.2 s recording (*Figure 3—video 1*). (**B, C**) The time courses of integrated intensities color-coded for $I_0$, $I_{45}$, $I_{90}$ and $I_{135}$ of two bifunctional-rhodamine-labeled AdiC molecules in the absence (**B**) or presence (**C**) of 0.75 mM $Arg^+$, from which $I_{tot}$ is calculated. Each vertical line in the black traces, superimposed on the colored traces, indicates the time point at which a change in the fluorophore's orientation is identified, whereas the horizontal lines represent the mean intensity between two identified consecutive time points. (**D–G**) The traces $\theta_L$ and $\varphi_L$ (**D, E**) in the laboratory frame of reference calculated from black intensity traces, which were rotated into the local frame of reference (**F, G**). (**H, I**) Values of $\Omega$ calculated according to *Equation 33* from the changes in either $\theta_L$ and $\varphi_L$ or $\theta$ and $\varphi$.

The online version of this article includes the following video, source data, and figure supplement(s) for figure 3:

*Figure 3 continued on next page*

*Figure 3 continued*

**Source data 1.** Intensity and angle data for [Arg]=0 mM condition.

**Source data 2.** Intensity and angle data for [Arg]=0.75 mM condition.

**Figure supplement 1.** Total intensities of individual molecules including the bleaching step.

**Figure supplement 1—source data 1.** Intensity data including the bleaching step for [Arg]=0 mM condition.

**Figure supplement 1—source data 2.** Intensity data including the bleaching step for [Arg]=0.75 mM condition.

**Figure 3—video 1.** A 7.2 second-long video of the four polarized emission intensities captured on an EMCCD camera, ordered top to bottom as $I_0$, $I_{45}$, $I_{90}$ and $I_{135}$.

https://elifesciences.org/articles/82175/figures#fig3video1

## Detection of intensity changes

As a protein molecule transitions from one conformation to another conformation, the orientation of the attached fluorophore changes with respect to the fixed polarization angles of the two polarized beam splitters. Consequently, the intensities of the four polarized fluorescence components undergo characteristic changes. For example, when the fluorophore's mean dipole vector moved such that its $\varphi$ angle increased from 0 to 90°, the intensity recorded in the 0° channel would decrease, whereas that in 90° would increase accordingly; the intensities of 45° and 135° channels would also characteristically change in directions opposite to each other. Compared to many other types of fluorescence-based methods, such as FRET, the expected concurrent, characteristic changes in all four channels here, which serve as a critical constraint in analysis, markedly increase the confidence of the detection of transitions, given that such types of changes are not expected for statistically random changes, namely, the so-called noise.

By examining the characteristic changes in the four fluorescence components with a so-called changepoint algorithm (*Chen and Gupta, 2001*), we detected the time point where a change in the fluorophore's orientation occurred, which was brought about by the underlying protein conformational change. To do so, we assessed the change in the number of recorded photons ($N$) per unit time with 95% confidence by evaluating the likelihood of two alternative possibilities that a change did or did not occur within a given time interval. This assessment was based on concurrent changes in the recorded $N$ among all four polarized components, redundancy that markedly increased the confidence that identified transitions were genuine.

During individual consecutive 10 ms recording intervals, the mean $N$ for all four polarization components together was 92 ('Materials and methods'), with an effective SNR of 7. As shown in *Figure 3B and C*, individual detected transitions are demarcated by the vertical lines in the black traces superimposed on the data traces. From the polarization properties of 92 photons on average, we could reliably detect the individual transitions in the fluorophore orientation among different conformational states at the intended time resolution. Thus, this method offers an exquisitely sensitive detection of changes in the fluorophore orientation.

## Angle calculation and state identification

Resolution of individual conformational states in terms of $\theta$ and $\varphi$ angles requires a higher SNR and thus a much greater number of photons than what is required for detecting the fluorophore's orientation changes. One way to solve this problem would be to increase the number of emitted photons by raising the intensity of the excitation laser, but a strong excitation intensity would undesirably shorten the lifetime of the fluorophore. An alternative way to increase SNR is to use the total number of photons recorded from each of the four channels within the duration of an event when an examined protein molecule adopts a specific conformation, dubbed dwell time. Practically, given that angles were calculated from ratios of intensities, a calculation using the total or the average number of photons recorded during an event would yield the same result.

Our ability to identify individual state-transition points was a prerequisite for us to perform such a summation, or averaging, of the number of recorded photons for a given event because individual dwell times were demarcated by these points. As shown in *Figure 3B and C*, while the vertical lines in the black traces, superimposed on the observed intensities traces, indicate the individual orientation-transition points, the average intensities over individual dwell times are shown as the horizontal lines in the black traces.

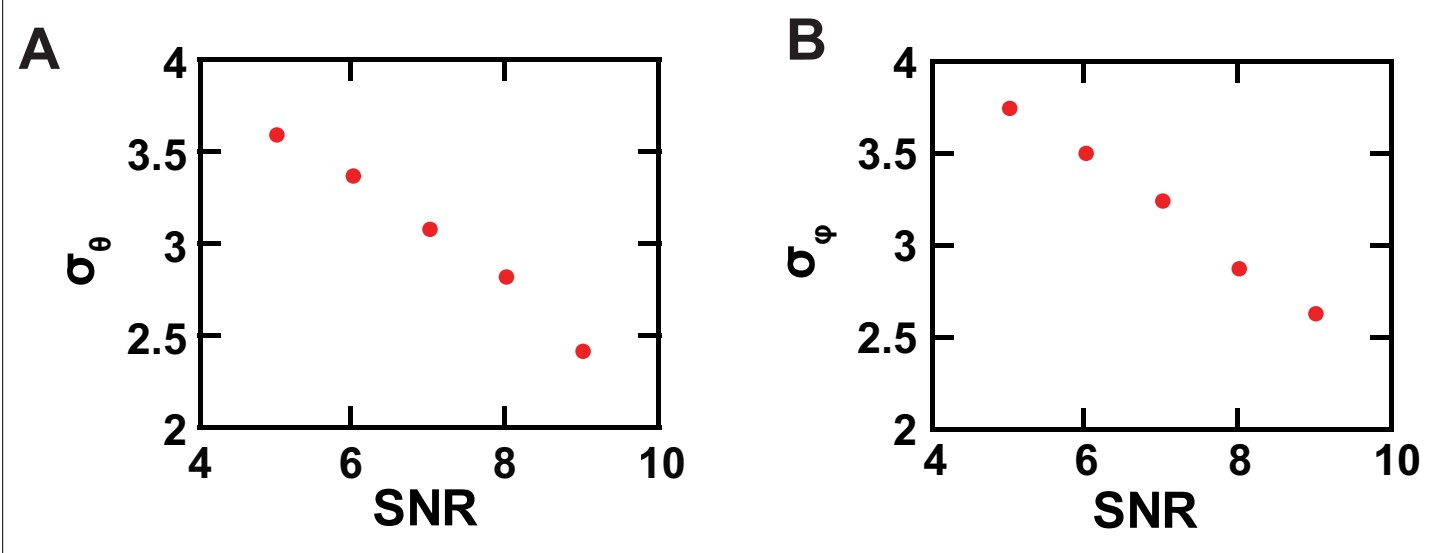

**Figure 4.** Experimental resolution of conformational states. (**A, B**) The $\sigma$ values of the $\theta$ (**A**) and $\varphi$ (**B**) populations, built with data acquired from individual single AdiC molecules labeled with bifunctional rhodamine, are plotted against SNR.

The online version of this article includes the following source data for figure 4:

**Source data 1.** Data for the relation between SNR and σ of φ and $\theta$ .

From these black traces of $I_0$ , $I_{90}$ , $I_{45}$, and $I_{135}$ , we calculated $\theta_L$ and $\varphi_L$ traces that are color coded (**Figure 3D and E**); these two angles are defined in the standard laboratory (L) frame of reference for microscopy studies where the z-axis is defined as being parallel to the optical axis of the objective and the x-y plane parallel to the sample coverslip. All angle calculations were done using expanded versions of **Equations 1 and 2** (**Equations 11 and 13**), which contain four necessary, predetermined system parameters. These previously established parameters numerically correct for incomplete photon collection ($\alpha$) (**Axelrod, 1979**), depolarization caused by imperfect extinction ratios of the polarizers (f) (**Lewis and Lu, 2019c**) and by fast wobble motion of the fluorophore dipole ($\delta$) (**Forkey et al., 2005**), and slightly different intensity-recording efficiency of the four polarization channels (g). In the angle calculation, we used the information from all photons (1740 photons on average) recorded from all four polarization channels during individual dwell times to determine the corresponding (mean) angles over these individual durations with adequately low $\sigma$. The relations between $\sigma$ for the two angles and SNR among individual analyzed particles are shown in **Figure 4**, in which the highest value of <4° is translated to a resolution of better than 10°.

Subsequently, conformational state populations could be identified from both angles together to increase resolvability and confidence, without any preconceived kinetic model. For the ease in technique development, in a previous study of the isolated gating ring of the MthK channel, the orientation of individual protein molecules was strictly constrained such that their central axis was well aligned with the optical axis. Under such a stringent condition and for that specific protein, its conformations could be resolved from the differences in the $\theta$ angle alone. As such, the conformational states could be directly sorted in terms of $\theta$ and $\varphi$, which is mathematically a 2D operation. Because one angle was adequate there, the sorting effectively became a 1D operation. Here, we need to rely on the combined differences on a 3D sphere. The asymmetric characteristics of $\theta$ and $\varphi$, illustrated below, would introduce some biases in state identification if these two angles were used in a 2D sorting operation. The angles $\theta$ and $\varphi$ are related to x,y,z coordinates in the corresponding Cartesian system as

$$\theta = arcsin\sqrt{\frac{x^2+y^2}{x^2+y^2+z^2}} \qquad (3)$$

and

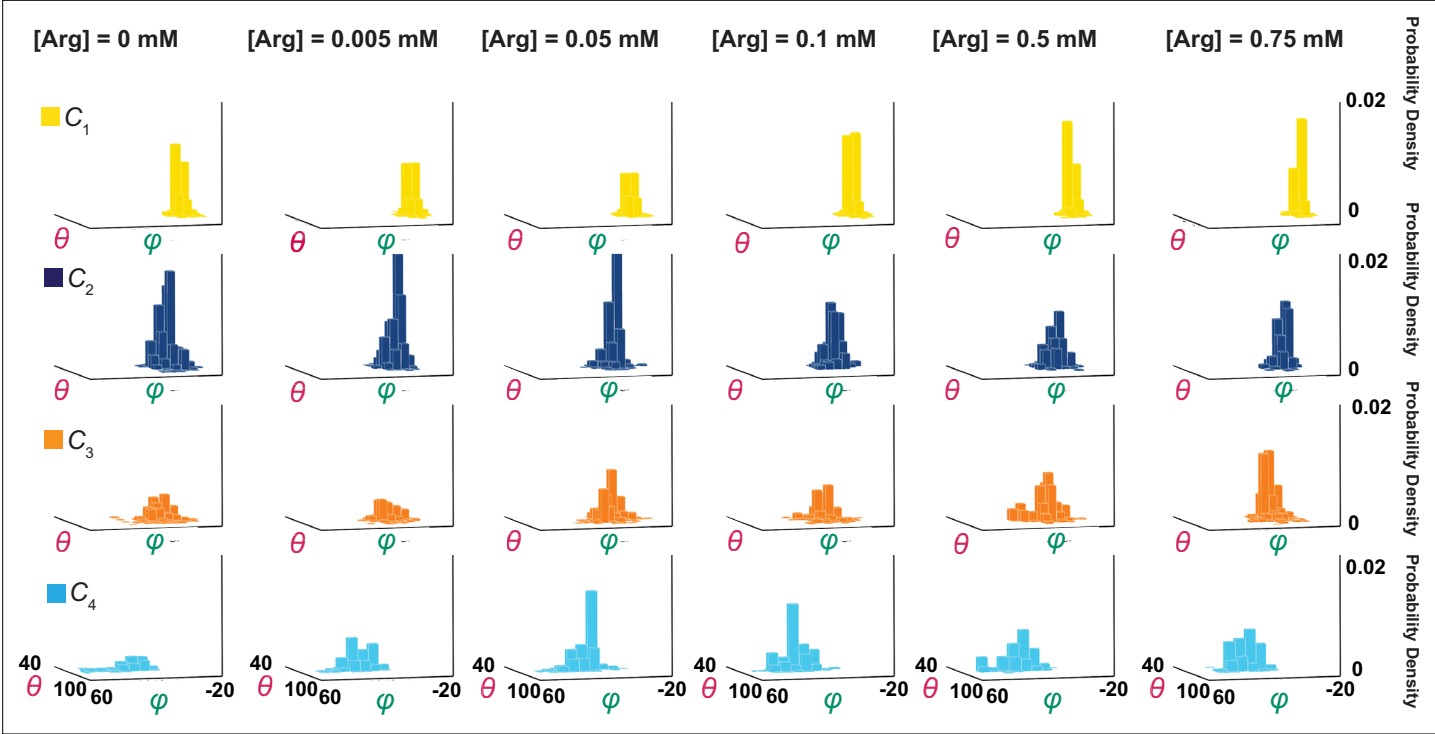

**Figure 5.** Ensemble 3D probability density distributions of $\theta$ and $\varphi$. The $\theta$ and $\varphi$ distributions of four individual states in the absence or presence of the indicated concentrations of Arg$^+$, in which the value of $\varphi$ is plotted along the x-axis, the value of $\theta$ along the y-axis, and the value of probability density along the z-axis. Distributions were built with the data analyzed from 91 or 75 number of particles with a total 3048 or 1494 number of events. Data columns for the conformational state $C_1$ are colored yellow, $C_2$ colored blue, $C_3$ colored orange, and $C_4$ color cyan.

The online version of this article includes the following source data for figure 5:

**Source data 1.** $\varphi$ angles for probability density distributions, organized by state and [Arg$^+$].

**Source data 2.** $\theta$ angles for probability density distributions, organized by state and [Arg$^+$].

**Source data 3.** $\varphi$ and $\theta$ sample histogram data for [Arg$^+$] = 0 mM.

**Source data 4.** $\varphi$ and $\theta$ sample histogram data for [Arg$^+$] = 0.75 mM.

$$\varphi = arctan\frac{y}{x} \tag{4}$$

where $\theta$ contains the information of all x, y, and z, whereas $\varphi$ contains that of only x and y. Furthermore, the two angles are related to x and y differently. To ensure equal weighting of the contributions from all three dimensions in the sorting operation, we mapped the information of the two angles on a unit sphere defined in a Cartesian coordinate system and performed the sorting in terms of x, y, and z coordinates on the basis of the shortest distance principle.

This 3D-sorting task would be extremely challenging for us to do manually on subjective visual guidance, if possible. Thus, to effectively sort individual events into distinct populations, we developed a *k*-means-clustering algorithm to achieve the solution for the shortest-distance case, which was optimized with two coupled algorithms (*simulated annealing* plus *Nelder–Mead downhill simplex*) (*Press et al., 2007*). The program examined from the case of a one-state distribution to the case of *k*-state distributions, one at a time in a series of separate operations. When two consecutive events were determined to belong to the same state distribution, the transition between them identified by the changepoint algorithm would most likely be part of the expected false-positive outcomes. These events would then be merged to form a single event for the state distribution. This operation reduces the false-positive identification of transition points.

For each successful operation with a specifically set number of states, the resolution criterion was ensured, that is, peaks of all populations were separated by at least 2.5σ. By this common criterion, the highest number of resolvable states was 4 over the examined Arg$^+$ concentration range (*Figures 5 and 6*); in these two figures, $\theta$ and $\varphi$ are the angle coordinates of a local framework transformed from

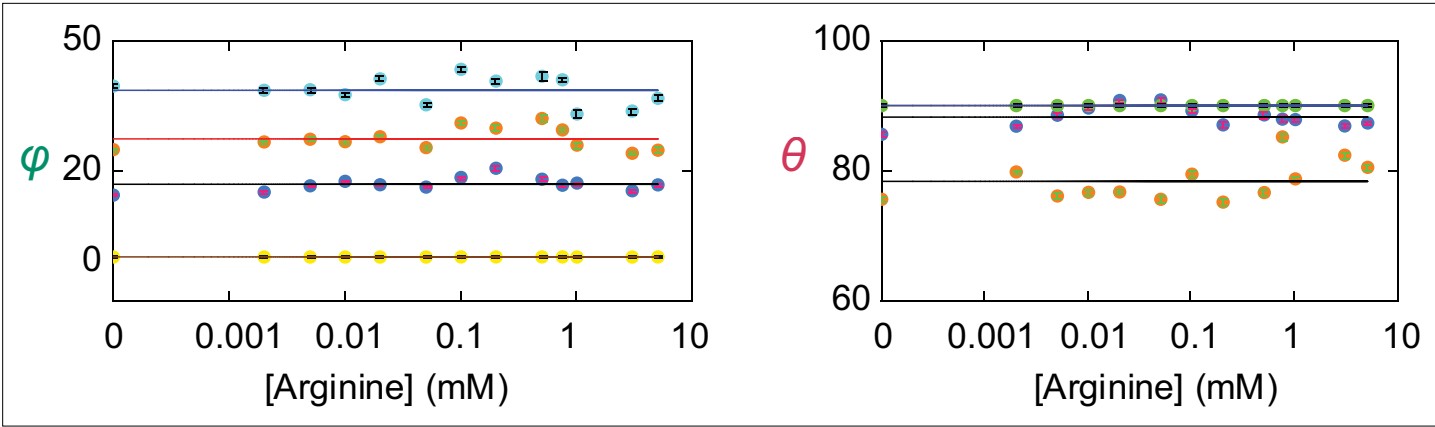

**Figure 6.** Angle values of individual conformational states. The values of $\theta$ and $\varphi$ (mean ± sem) for each of the four conformations are plotted against the concentration of Arg$^+$. The number of events is 691–3084. The symbols for the conformational state $C_1$ is colored yellow, $C_2$ colored blue, $C_3$ colored orange, and $C_4$ color cyan. Note that $\theta$ for both $C_1$ (yellow) and $C_4$ (cyan) are set to 90° as described in the text and are thus overlapped.

The online version of this article includes the following source data for figure 6:

**Source data 1.** Table of mean values for $\theta$ and φ organized by state and [Arg$^+$].

$\theta_L$ and $\varphi_L$ of the standard laboratory framework, as described below. The conformational states are numerated according to the sequence in the solution to achieve the shortest pathway among the four states, denoted as $C_1$, $C_2$, $C_3$, and $C_4$ ('Materials and methods'). To illustrate the sorting result, x, y, and z coordinates of individual events in the Cartesian system or their angle coordinates in a spherical system are presented in *Figure 7*, in which they are color-coded according to states. Thus, the minimal model must have four states for either the apo or the substrate-bound forms, totaling eight states.

## Spatial alignment of individual molecules

For greater likelihood to accurately estimate angle values, we need to determine their mean values from individual events of numerous molecules that had already been analyzed individually for necessary precision of angle measurements and thus spatial resolution. This operation requires all molecules be aligned in the same orientation. Unfortunately, individual molecules on the coverslip, and consequently the tracked helix, were randomly oriented in the x-y plane relative to the *x*-axis, that is, their $\varphi_L$ varying randomly among different molecules. This problem previously prevented us from building the distribution of $\varphi_L$ for a given state among individual molecules (*Lewis and Lu, 2019c*). Furthermore, each dimer molecule of AdiC is anchored to the cover slip coated with streptavidin through two available biotin moieties and four streptavidin-binding tags, each of which was covalently linked to an N- or C-terminus of the polypeptides of two subunits. As mentioned above, these terminal binding regions were of some flexibility. Consequently, the twofold symmetry axis of the individual dimer molecules was not aligned with the optical (z) axis of the microscope framework. As such, the orientation of one molecule varied considerably from that of another (e.g., *Figure 7A* versus *Figure 7B* or *Figure 7E* versus *Figure 7F*). Resolving this issue would also pave the way for studying individual randomly oriented molecules in the future.

To align all the molecules during analysis, we mathematically rotate them from the laboratory frame of reference into a local coordinate system defined on the basis of the spatial features of the tracked helix in the protein (*Figure 7—figure supplement 1A*). This system is defined such that the tracked helix in $C_1$ is always aligned with the local x-axis, that is, mean $\varphi_1$=0°, and the helix in the plane defined by $C_1$ and $C_4$ is always in the local x-y plane, that is, mean $\theta_1$ or $\theta_4$=90°. These two features fully define the x,y,z-axes, and thus $\theta$ and $\varphi$ in the local framework of AdiC. Evaluated in this local framework (*Figures 3F, G, 7C, D, G and H*), individual events of the same state observed with all molecules under a given condition could be used to build a single distribution (*Figure 5*). From the distribution for each state, we determined the mean $\theta_i$ and $\varphi_i$. To specify the six mean direct-angle $\Omega_{i,j}$ values among the tracked dipole orientations in the four states, which were calculated using *Equation 33*, we used four unit-vectors to specify the orientations in the local framework (*Figure 8A*). The mean $\theta_i$, $\varphi_i$, and $\Omega_{i,j}$ values are all summarized in *Figure 8B and C*.

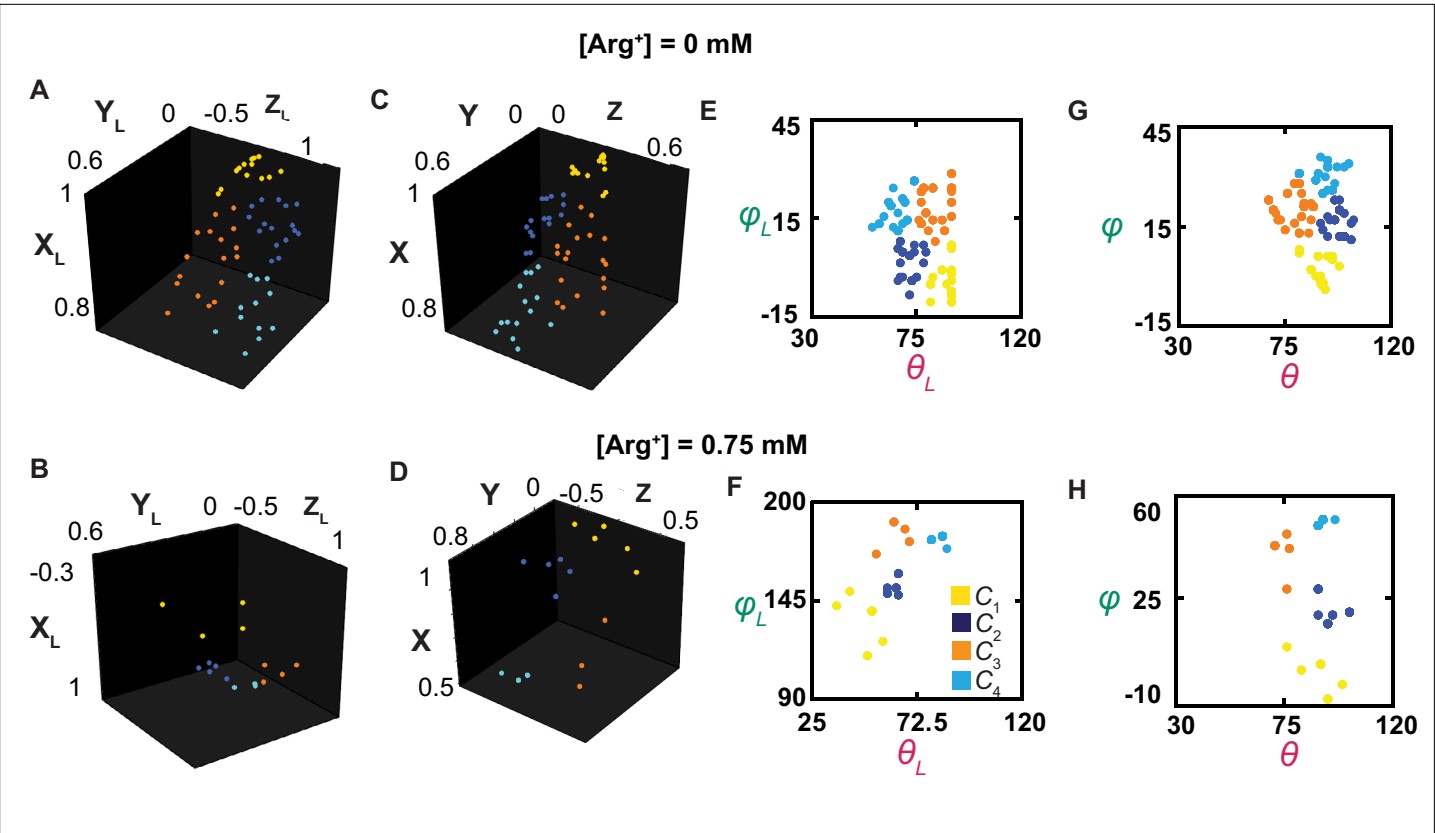

**Figure 7.** The orientations of the dipole vector of the fluorescence probe in different conformational states. (**A–D**) The positions of the arrowheads of individual dipole vectors are mapped onto a unit sphere, on the basis of being in a Cartesian coordinate system defined according to the usual laboratory framework (**A, B**) or a local one (**C, D**), described in the text. The $X_L$, $Y_L$, and $Z_L$ positions calculated from $\theta_L$ and $\varphi_L$, determined in *Figure 3*, and a radius of a unit length. (**E–H**) The inclination and rotation angle values are plotted against each other in the laboratory (**E, F**) or the local (**G, H**) frame of reference. The data points for conformational state $C_1$ are colored yellow, $C_2$ colored blue, $C_3$ colored orange, and $C_4$ color cyan.

The online version of this article includes the following source data and figure supplement(s) for figure 7:

**Source data 1.** Orientation information of the fluorophore for [Arg] = 0 mM.

**Source data 2.** Orientation information of the fluorophore for [Arg] = 0.75 mM.

**Figure supplement 1.** Transformation between the laboratory and local coordinates, and assignment of states.

Following the rotation operation, we could also build a distribution of mean angle values of individual molecules to illustrate the molecule-to-molecule variation in $\theta_i$ and $\varphi_i$ (*Figure 9*). If the contribution of each molecule were weighted by the length of the trace, the mean value of each distribution in *Figure 9* should statistically be the same as the corresponding one in *Figure 5*, built with the values of individual events from all molecules. Although the ensemble mean angle values should be more accurate statistically, the apparent angle resolution would be poorer. Indeed, compared with $\sigma$ for individual particles (*Figure 4*), the ensemble angle distributions built with data determined from molecule-by-molecule analyses have much larger $\sigma$ (8–11°, *Figures 5 and 8B*), except for those of $\varphi_1$ and $\theta_1$ or $\theta_4$ which are normalized such that their mean values equal 0° and 90°. Thus, the initial molecule-by-molecule data analysis, as shown in *Figure 3*, was essential for resolving individual conformational states in terms of $\theta$ and $\varphi$.

## Determination of equilibrium constants among the Apo and Arg$^+$-bound conformational states

As an essential evaluation of the basic energetic properties of the protein's individual states, we examined individual AdiC molecules in the presence of a large series of Arg$^+$ concentrations. Both sides of an AdiC molecule facing the same solution allowed us to determine the probabilities of individual conformational states under conditions where the system as a whole was in equilibrium, through

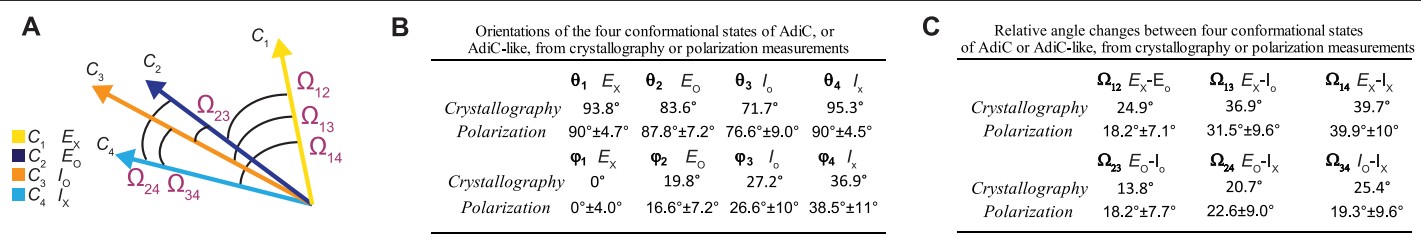

**Figure 8.** Comparison of the orientations of helix 6A in the corresponding states determined from the crystal structures and in the polarization study. (**A**) Depiction of the six $\Omega$ angles among the four orientations of the helix in the four states, represented by four arrows color-coded for states. (**B, C**) The $\theta$ and $\varphi$ angles (**B**) of the helix for corresponding states determined from the crystal structures and in the polarization study, as well as the $\Omega$ angles (**C**), are compared in the local frame of reference. All angle values for the conformational states determined by polarization are presented as mean ± standard deviation ($\sigma$). In the structural analysis, the AdiC structures in the states $E_O$ (blue) (PDB: 7O82) and $E_X$ (yellow) (PDB: 3L1L), and the BasC and ApcT structures in the states $I_O$ (orange) (PDB: 6F2G) and $I_X$ (cyan)(PDB: 3GIA) were used.

The online version of this article includes the following source data for figure 8:

**Source data 1.** $\theta$ and $\varphi$ values of the four conformational states.

**Source data 2.** $\Omega$ values of the four conformational states.

which we could straightforwardly obtain equilibrium constants ($K_{ij}$) among the states, including dissociation constants ($K_{Di}$) to be compared in 'Discussion' with previously reported $K_D$ values.

Shown in *Figure 5* are plots of the probability density distributions of $\theta_i$ and $\varphi_i$ of each of the four states in a series of Arg$^+$ concentrations, built with the data from numerous individually analyzed

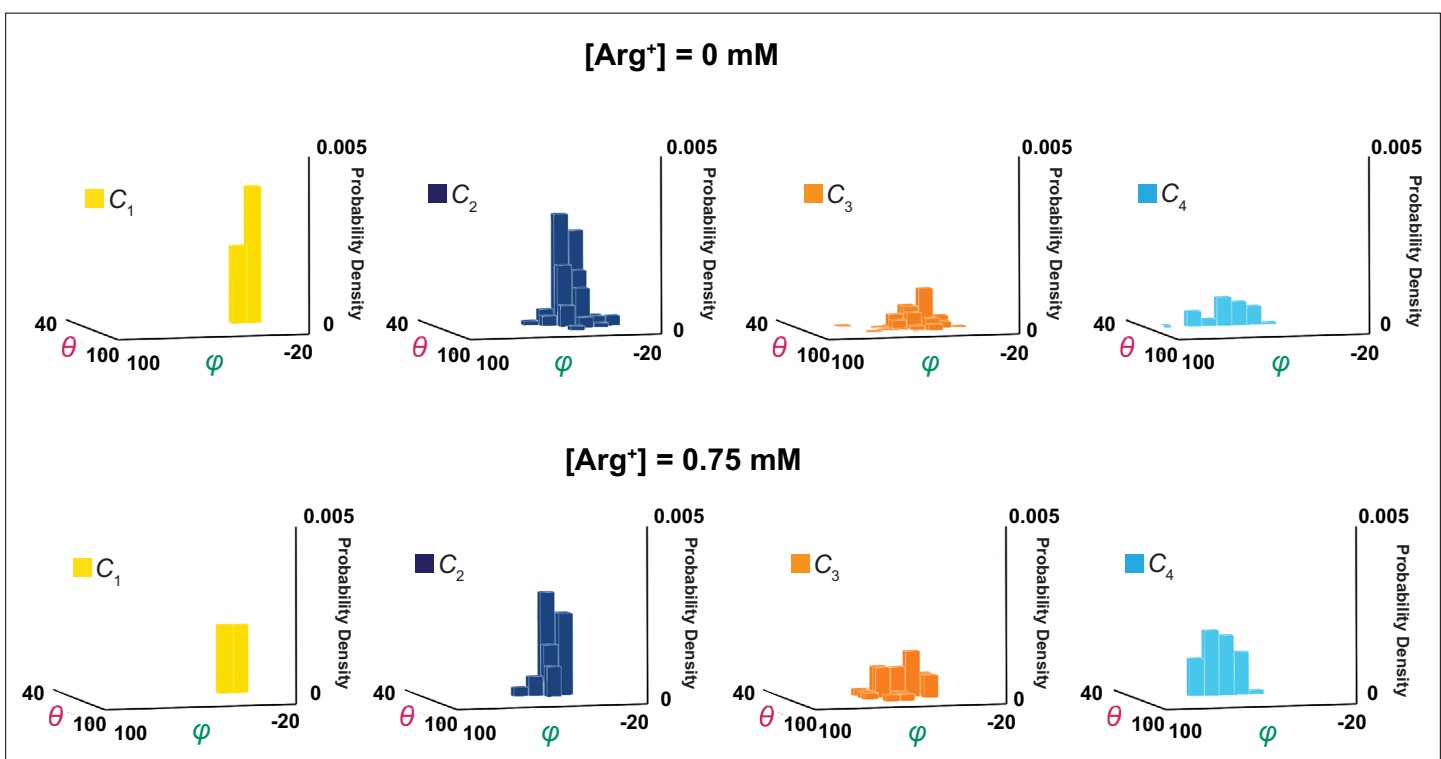

**Figure 9.** 3D probability density distributions of mean $\theta$ and $\varphi$ determined from individual molecules. The value of $\varphi$ is plotted along the x-axis, the value of $\theta$ along the y-axis, and the value of probability density along the z-axis, obtained in the absence (top) and presence (bottom) of a saturating concentration of Arg$^+$. Distributions were built with the data analyzed from 34 or 91 number of particles with a total 691 or 3048 number of events. Data column for the conformational state $C_1$ is colored yellow, $C_2$ colored blue, $C_3$ colored orange, and $C_4$ color cyan.

The online version of this article includes the following source data for figure 9:

**Source data 1.** $\varphi$ data for mean angle distributions.

**Source data 2.** $\theta$ data for mean angle distributions.

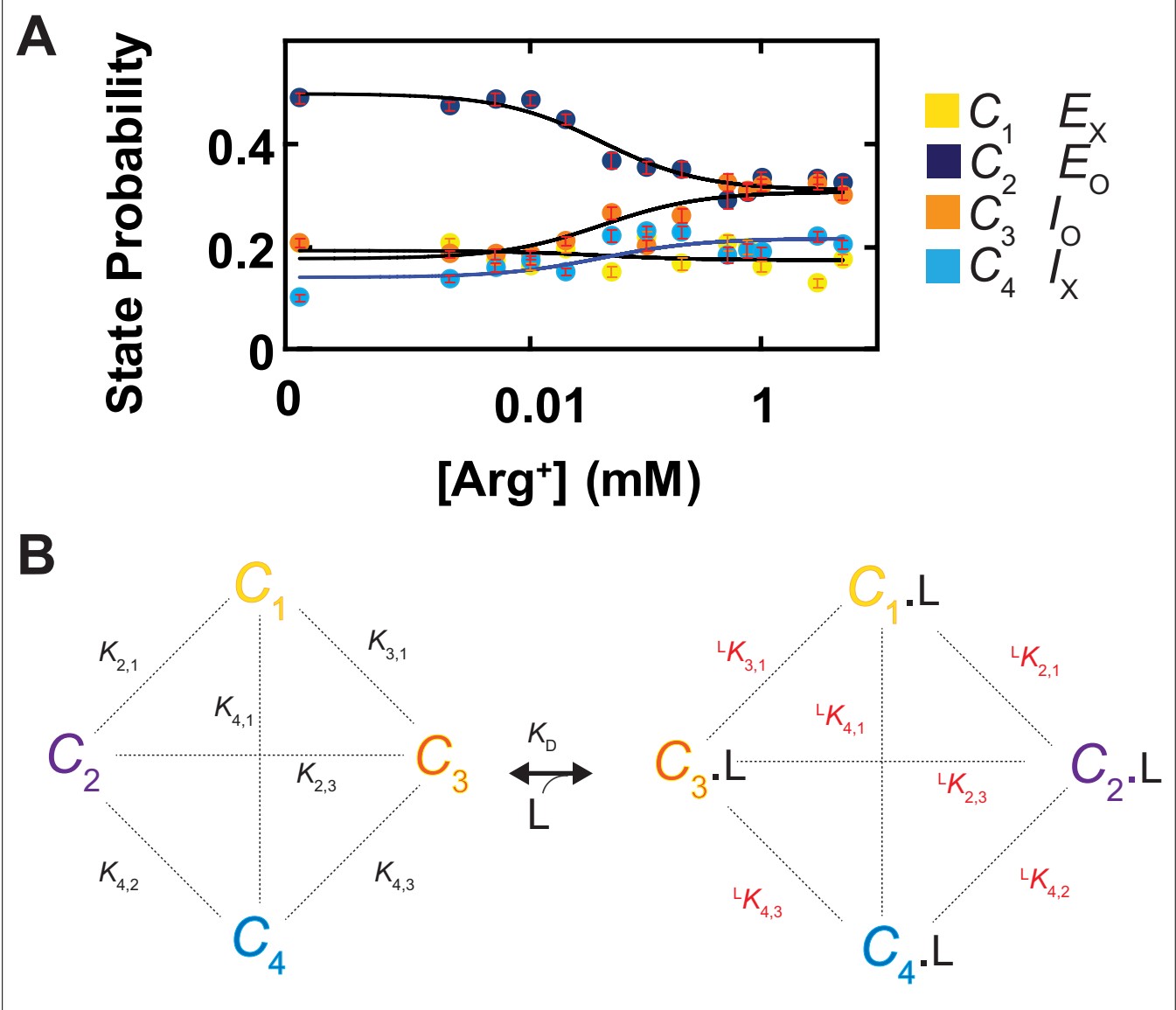

**Figure 10.** Ligand dependence of the probabilities of conformational states and the diagram of a conformational state model of AdiC. (**A**) The probabilities of individual states (mean ± sem, number of events is 691–3084) are plotted against the Arg⁺ concentration on a logarithm scale. The four curves superimposed on the data correspond to a global fit of a model in which the interaction between the subunit of AdiC and Arg⁺ has an one-to-one stoichiometry. The fitted values of all parameters are summarized in *Table 1*. (**B**) An eight-state model that accounts for the observed conformational behaviors of AdiC: four apo states and four ligand-bound states.

The online version of this article includes the following source data for figure 10:

**Source data 1.** State probabilities and associated errors organized according to [Arg⁺].

particles. These angle distributions contain the information regarding the probabilities of occupying each of the four states in the presence of the corresponding Arg⁺ concentrations (*Figure 10A*). All four states appeared in the absence and the presence of Arg⁺. With increasing concentration, the probability of $C_3$ increased whereas that of $C_2$ decreased. In contrast, the probability of $C_1$ or $C_4$ exhibited relatively small changes. Nonetheless, these observations indicate that all states bind Arg⁺ because any state that could not bind Arg⁺ would practically vanish in a saturating concentration of Arg⁺. The observation of all four conformations in the absence or the presence of Arg⁺ is consistent with the Monod–Wyman–Changeux model for ligand-dependent conformational mechanism of allosteric proteins in that all conformations of a protein occur in the absence of ligand, and the binding

**Table 1.** Probabilities and equilibrium constants for apo and Arg$^+$ bound states.

**Apo state**

|  |  | $P_1$ | 0.192 + 0.015/–0.024 |  |  |
| --- | --- | --- | --- | --- | --- |
| $K_{2,1}$ | 2.566 + 0.589/–0.244 | $P_2$ | 0.492 + 0.040/–0.014 |  |  |
| $K_{3,1}$ | 0.923 + 0.061/–0.149 | $P_3$ | 0.177 + 0.011/–0.037 |  |  |
| $K_{4,1}$ | 0.729 + 0.225/–0.103 | $P_4$ | 0.140 + 0.026/–0.014 |  |  |
| **Arg$^+$ bound state** |  |  |  | **K$_D$ (µM)** |  |
|  |  | $^{arg}P_1$ | 0.172 + 0.020/–0.020 | $^{arg}K_{D1}$ | 49 + 46/–25 |
| $^{arg}K_{2,1}$ | 1.795 + 0.298/–0.319 | $^{arg}P_2$ | 0.308 + 0.013/–0.025 | $^{arg}K_{D2}$ | 69 + 67/–33 |
| $^{arg}K_{3,1}$ | 1.782 + 0.272/–0.193 | $^{arg}P_3$ | 0.306 + 0.019/–0.018 | $^{arg}K_{D3}$ | 25 + 20/–14 |
| $^{arg}K_{4,1}$ | 1.250 + 0.228/–0.188 | $^{arg}P_4$ | 0.214 + 0.018/–0.015 | $^{arg}K_{D4}$ | 28 + 31/–14 |

Parameters from fit of **Equation 5** to the plots of AdiC conformational state probabilities versus [Arg$^+$] (**Figure 10A**). 95% confidence intervals were calculated from the maximum likelihood optimization of 1000 sets of simulated data obtained from bootstrapping.

of a ligand does not create a new conformation but merely energetically stabilizes a spontaneous conformation that is ready to capture the ligand (**Monod et al., 1965**).

Here, a model with a minimum number of eight states is required to account for the observed conformational behavior of AdiC: a set of four without ligand bound ($C_1$, $C_2$, $C_3$, and $C_4$) and another set of four with the Arg$^+$ ligand (L) bound ($C_1$.L, $C_2$.L, $C_3$.L, and $C_4$.L) (**Figure 10B**). The probability $p_i$ of occupying state $i$ is expressed as

$$p_i = \frac{K_{i,1} + {}^L K_{i,1}\frac{[\text{L}]}{K_{D1}}}{\sum_{i=1}^{4}K_{i,1} + \sum_{i=1}^{4}{}^L K_{i,1}\frac{[\text{L}]}{K_{D1}}} \tag{5}$$

where $C_1$ is used as a reference for the other states ($C_i$) to define equilibrium constants:

$$K_{i,1} = \frac{[C_i]}{[C_1]}; \quad {}^L K_{i,1} = \frac{[C_i.L]}{[C_1.L]}; \quad K_{Di} = \frac{[C_i][L]}{[C_i.L]}; \quad i = 1, 2, 3, 4 \tag{6}$$

where $K_{1,1}$ is defined to equal one. Each equilibrium constant reflects the free energy difference between a pair of states that may or may not be connected kinetically. In principle, the equilibrium constants for the apo or Arg$^+$-bound states are constrained by the state probabilities under the condition of zero or saturating Arg$^+$, and $K_{Di}$ by the so-called midpoint positions of the curves (**Figure 10A**). In practice, we determined these parameters by fitting **Equation 5** to the four plots in **Figure 10A** simultaneously as a global fit, all summarized in **Table 1**. Together, these constants would fully define the energetic relations among the eight states. If needed, the remaining six equilibrium constants ($K_{3,2}$, $K_{4,2}$, $K_{4,3}$, $^L K_{3,2}$, $^L K_{4,2}$, and $^L K_{4,3}$) could be calculated from the ones given in **Table 1**. Note that even though the concentration-dependent plot of the probability of $C_1$ or $C_4$ is relatively flat, their $K_D$ values are determinable. This is because the model can be fully specified by a certain combination of merely seven equilibrium constants, e.g., three of the six $K_{i,1}$ values, three of the six $^L K_{i,1}$ values, and only one of the four $K_D$ values. As such, in the present so-called overdetermined case, the $K_D$ values of the relatively flat traces are fully constrained by those more curved traces in a global fit to all four plots simultaneously.

## Discussion

In this study, on the basis of $\theta$ or $\varphi$ alone, the smallest angle changes resolved, namely, those between $C_2$ and $C_3$, are ≥10° (**Figure 8B**), which requires a σ ≤ 4°. Thus, it is justified to use ultimately the intensity data with SNR of ≥ 5, which corresponds to σ for the $\theta$ and $\varphi$ distributions of < 4°, translated to a minimum resolution of better than 10° (**Figure 4**). Note that our actual use of the combined information of $\theta$ and $\varphi$ should lead to higher effective resolution.

The present examination of the properties of conformational changes was performed under equilibrium conditions, which allowed us to determine straightforwardly the $K_D$ values of four conformations

from the dependence of their probabilities on the ligand concentration. The resulting values of $K_{D1}$ through $K_{D4}$ ranges from 25 to 69 μM (*Table 1*), statistically comparable with the previously reported overall $K_D$ determined by ITC: a relatively narrow range of 32–93 μM for one lab without lab-to-lab variations (*Fang et al., 2007*; *Tsai et al., 2012*), or a wider range of 32–204 μM when data from other labs are included (*Casagrande et al., 2008*; *Gao et al., 2010*; *Wang et al., 2014*). Thus, the $K_D$ values estimated by the present method are valid. Maneuvers such as introducing cysteine mutations and attaching the fluorophore to the AdiC molecule, which are part of what contributes to system errors of the present method, have no markedly consequential energetic impacts on its affinity for Arg$^+$. Such a finding is not particularly surprising because the chosen labeling part is on the surface of the protein such that it is not at, but external to, the ligand-binding site. Thus, as an advantage of the present method, one can and should choose to attach the label to the surface of a moving part of the protein, a part that is not of functional activity.

Generally, a transporter is expected to adopt four main types of structure-function states in terms of the accessibility of its external and internal sides to substrates, dubbed the externally open ($E_o$), externally occluded ($E_x$), internally open ($I_o$), and internally occluded ($I_x$) states (*Post et al., 1972*; *Gao et al., 2010*; *Krammer and Prévost, 2019*). By this definition, when a transport molecule adopts the $E_o$ and $I_o$ states, it is accessible only to extracellular and intracellular ligands, respectively. The molecule in the $E_x$ or $I_x$ state is inaccessible to ligands from either side. Thus far, the crystal structures of AdiC in the $E_o$ and $E_x$ state have been solved (*Gao et al., 2009*; *Fang et al., 2009*; *Gao et al., 2010*; *Kowalczyk et al., 2011*; *Ilgü et al., 2016*, *Ilgü et al., 2021*). Additionally, the structures of BasC and ApcT (*Shaffer et al., 2009*; *Errasti-Murugarren et al., 2019*), which share the same fold with AdiC, were solved in $I_o$ and $I_x$ states, respectively.

Intriguingly, as shown in the results, the present study of fluorescence polarization resolved four macroscopic conformational-state populations, which was performed without any preconceived number of states. The probabilities of $C_2$ and $C_3$ clearly vary with the concentration of Arg$^+$, and these two states should thus be open states (*Figure 10A*). In contrast, the probabilities of $C_1$ and $C_4$ vary little, and thus are not open states directly accessible to ligands, which are consistent with occluded states in adopting the prevailing nomenclature. Under the present conditions, the comparable probabilities of $C_2$ and $C_3$ in a practically saturating concentration of Arg$^+$ are consistent with the following experimental findings obtained from previous flux assays: both sides of AdiC have comparable $K_m$, and the apparent maximal net flux rates of radioactive Arg$^+$, which were separately measured for the two opposite directions, are also comparable (*Tsai et al., 2012*; confer *Krammer et al., 2016*).

In the previous study of the isolated gating ring of the MthK channel, we could easily relate the states identified in the polarization study to the structural states on the basis of the matching relations of $\theta$ angles alone, which is effectively an 1D operation. For ease of communication, the former types of state will be referred to as the conformational states. However, to relate two sets of structural and conformational states in a 3D operation would require a quantitative approach. Thus, as an additional exercise, we illustrate below such an approach that we have developed.

Thus far, only the structures of the $E_o$ and $E_x$ states of AdiC have been solved, so for this exercise we used the structures of the transporters BasC and ApcT in the $I_o$ and $I_x$ states as proxies for those of AdiC because they all have the same structural fold (*Shaffer et al., 2009*; *Gao et al., 2010*; *Errasti-Murugarren et al., 2019*; *Ilgü et al., 2021*). This exercise can be readily repeated when the $I_o$ and $I_x$ structures of AdiC become available. The helix 6 in the two AdiC structures and that of its counterparts in BasC and ApcT adopts different orientations. To illustrate these differences, the four structures are aligned spatially, and their relevant portions are shown in *Figure 11A and B*. The orientation of the helix 6 or its counterpart is represented by a vector that is color-coded for a specific structural state (open arrow heads, *Figure 11C*). There are 24 possible ways of relating the four structural states to the four conformational states. To determine the best matching relation, we fit the unit-vectors (open heads, *Figure 11C*) and consequently their $\theta$ and $\varphi$ (open circles, *Figure 11D*), which specify the helix's orientations in the four structural states, to those of the conformational states in the local framework (closed heads or circles), while the internal spatial relations among four states in either set of compared data were fixed. To our surprise, in the statistically best matching case (*Figure 11C*), the four mean orientations of the fluorophore attached to the helix 6 apparently well match the orientations of this helix in the structures of AdiC, BasC, and ApcT captured in four different states (*Figure 8B and C*).

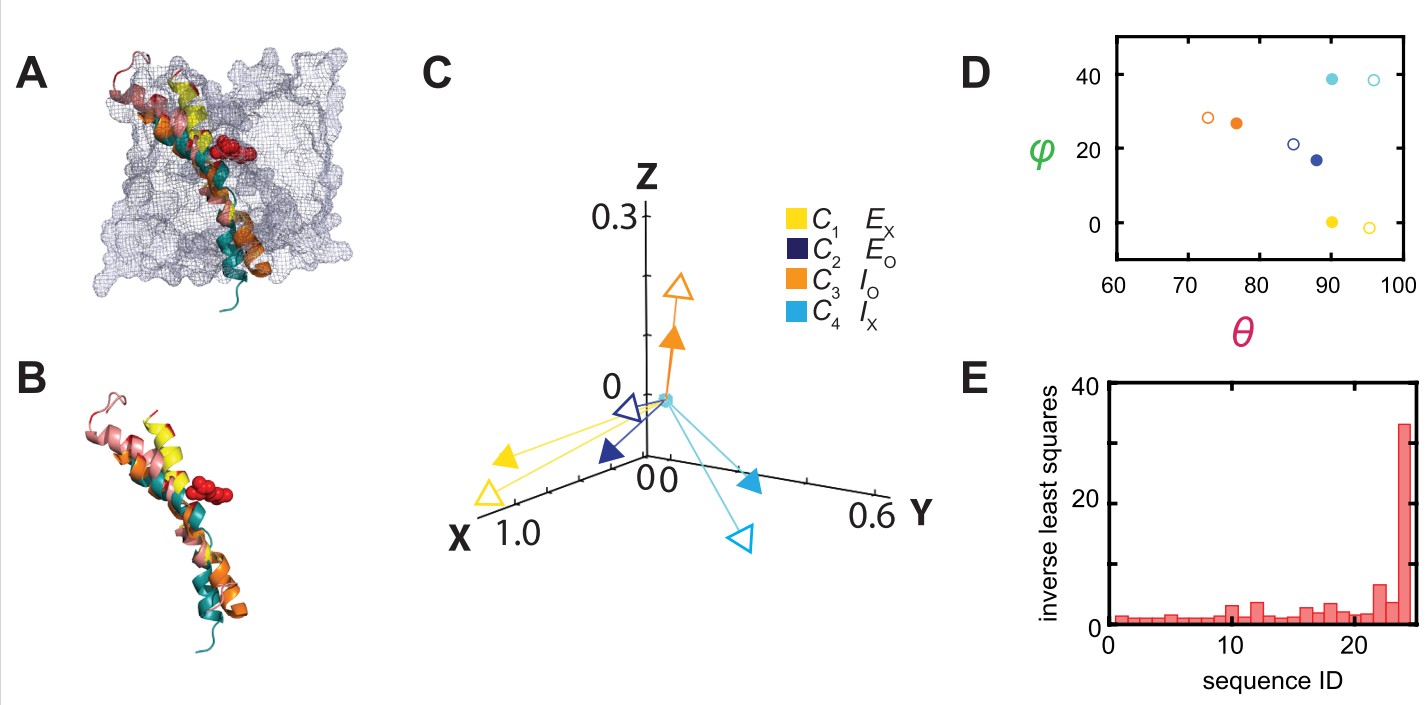

**Figure 11.** Relations between the structural and conformational states determined respectively from the crystal structures and in the polarization study. (**A, B**) Alignments of AdiC's helix 6A in the structural states $E_O$ (blue) (PDB: 7O82) and $E_X$ (yellow) (PDB: 3L1L) with the corresponding helices of BasC and ApcT in the states $I_O$ (orange) (PDB: 6F2G) and $I_X$ (cyan) (PDB: 3GIA), with (**A**) or without (**B**) the rest of the protein represented by a mesh contour. (**C**) The four mean orientations of the helix in the four conformational states are represented by a set of four unit-vectors (closed heads) in the local framework of coordinates whereas those for the four structural states by another set of unit vectors (open heads). The vectors for the conformational states are drawn according to mean $\theta$ and $\varphi$ values obtained from their respective distributions and color-coded for the corresponding states. The two sets of vectors are overlaid as described in the text. (**D**) Scatter plots of mean $\theta$ versus $\varphi$ values (closed circles) for four conformational states, which are compared with those for the four structural states (open circles); all are color coded for states. (**E**) The inverse values of combined least-distance-squares between the locations of the arrow heads of the two compared groups (open versus closed) in (**C**) for all 24 possible combinations among them.

The online version of this article includes the following source data for figure 11:

**Source data 1.** Tables comparing mean angles calculated from polarization measurements and those from structure.

For further illustration, we calculated the combined least-square ($LS_c$) values between the two sets of open and closed arrowheads in *Figure 11C* for all 24 combinations, and plotted the $\frac{1}{LS_c}$ values in *Figure 11E*. In the statistically best matching case based on the largest $\frac{1}{LS_c}$, $C_1$ would correspond to $E_x$, $C_2$ to $E_o$, $C_3$ to $I_o$, and $C_4$ to $I_x$, in which the values $\theta_i$ and $\varphi_i$ and $\Omega$ are comparable between the corresponding conformational and structural states (*Figure 8B and C*). In terms of open versus occluded states, the assignments of $C_2$ and $C_3$ as open states, and $C_1$ and $C_4$ as occluded states are consistent with those based on the Arg$^+$ concentration dependence, discussed above. In terms of sidedness, judging from the structures of AdiC, that is, $E_o$ and $E_x$, the spatial orientations of helix 6A in them ($\theta$ and $\varphi$) are compatible with those of $C_2$ and $C_1$.

In summary, using a state-of-the-art fluorescence polarization microscopy system assembled in house, we have, for the first time, successfully tracked conformational changes in a single integral membrane protein molecule among four states that occur on angstrom and millisecond scales. The resolving power of this technique allowed us to experimentally establish a fully determined quantitative model of eight states required to account for the energetics of the observed conformational changes of an AdiC transport molecule with or without a ligand bound. This capability of resolving and tracking the conformational states of individual molecules forms the foundation for performing the kinetic study to acquire the necessary dynamic information for understanding the transporting mechanism, and for creating an experiment-based 4D model to quantitatively account for the complex spatiotemporal behaviors of a transporter molecule, which can be applied to the investigation of other membrane proteins.

# Materials and methods

## Materials

Detergent n-dodecyl-β-D-maltopyranoside (DDM) was purchased from Anatrace, 1-palmitoyl-2-ol eoyl-glycero-3-phosphocholine (POPC) from Avanti Polar Lipid Inc, bifunctional rhodamine bis-((N-iodoacetyl)-piperazinyl)-sulfonerhodamine from Invitrogen (B10621), strep-tactin resin from IBA, and cover slip (#1.5) and microscope slide glass from Fisher Scientific or VWR. Unless specified otherwise, all other reagents were purchased from Sigma, Thermo Fisher Scientific, or EMD Millipore.

## Cloning, protein expression, and purification

A double-stranded DNA fragment, synthesized by Integrated DNA Technologies (IDT), contains, from the N-terminus to the C-terminus, Avi tag for recognition by biotin ligase, a Strep II tag, a linker (GGGSGGGS), the gene of AdiC of *Escherichia coli*, a linker (GGGS), a thrombin protease recognition site, a C-termini Strep II tag, and a stop codon, which was cloned into the pET28b vector. The removal of two native cysteines (C238A and C281A) and introduction of the double G188C and S195C cysteine mutations in helix 6a for attaching bifunctional rhodamine were carried out using the Quick-Change technique and verified by DNA sequencing.

The AdiC protein was expressed in *E. coli* BL21(DE3) cells transformed with AdiC-gene-containing plasmids. The transformed cells were grown in Luria Broth at 37°C to an $A_{600}$ of ~1.0. Protein expression was induced with 0.5 mM iso-propyl β-D-thiogalactopyranoside (IPTG) at 22°C overnight. The cells were harvested and resuspended in a solution containing 100 mM NaCl, 50 mM tris-(hydroxymethyl)-aminomethane (Tris) titrated to pH 8.0, 1 mM phenylmethylsulfonyl fluoride (PMSF), 4 mM tris-(2-carboxyethyl)-phosphine (TCEP), 1 µg/ml leupeptin, and 1 µg/ml pepstatin A. To extract membrane proteins, 40 mM DDM was added to the cell suspension; the conical tube containing this suspension was placed on a rotating device in a 4°C cold room for 2–3 hr. The cell suspension was then sonicated, the cell lysate was centrifuged at $12,000 \times g$ for 20 min, and the resulting supernatant was loaded onto a gravity flow column packed with strep-tactin super-flow resin (IBA). The column was washed with a wash buffer (WB) containing 100 mM NaCl, 50 mM Tris titrated to pH 8.0, 2 mM TCEP, and 2 mM DDM. AdiC protein was eluted from the column using WB added with 10 mM desthiobiotin. The AdiC-containing fractions were pooled and then concentrated with an Amicon Ultra concentrator (50K MWCO) before a further purification with a size-exclusion FPLC column (superdex 200 10/30, GE) equilibrated in WB.

## Labeling and reconstitution of AdiC into nanodiscs

Purified AdiC was mixed with POPC and the MSP2N2 membrane scaffold protein purified as described (*Ritchie et al., 2009*; *Denisov et al., 2019*) in an 1:1500:10 molar ratio of the AdiC dimer:POPC:MSP2N2. The mixture was incubated at 4°C for at least 2 hr. Nanodiscs were assembled during a dialysis of the mixture against a buffer containing 100 mM NaCl, 20 mM Tris titrated to pH 8.0, and 0.5 mM TCEP at 4°C overnight. After the dialysis, the nanodiscs were labeled with biotin using the BirA Enzyme (Avidity) following the protocol provided by the manufacturer, and then mixed with bifunctional rhodamine in a 2:1 molar ratio of the AdiC dimer:bifunctional rhodamine, a ratio to maximize the chance of only one AdiC monomer in each dimer to be labeled; this mixture was incubated at room temperature for more than 4 hr. The remaining free dye was removed first with Biobeads (SM-2, Bio-Rad) and then by size-exclusion chromatography (Superose 6 10/30 or Superderx 200 10/300, GE) equilibrated with a solution containing 100 mM NaCl and 50 mM Tris titrated to pH 8.0. As expected, the peak of the protein containing the two mutant cysteine residues, detected at 280 nm, co-migrated with that of the fluorophore dye detected at 550 nm. In contrast, no notable absorbance peak at 550 nm co-migrated with the peak of the protein without these mutant cysteine residues. Thus, there was no detectable background labeling (*Figure 1—figure supplement 1A and B*). The final product of nanodiscs harboring labeled AdiC was aliquoted, flash frozen in liquid nitrogen, and stored in a –80°C freezer.

## Sample preparation for data collection with polarization TIRF microscope

For adhesion of streptavidin, one side of a cover slip is exposed to 0.01% poly-L-lysine solution for 1 hr, before being rinsed with distilled water and air dried. Prior to an experiment, a cover slip was attached, via thin transparent Scotch adhesive tapes placed on its left and right edges, to a microscopy slide, with the poly-L-lysine-coated side of the cover slip facing the bottom side of the slide. Solutions were to be placed in the space between the two pieces of glass created by the adhesive tapes that acted as a spacer. The poly-L-lysine-coated side of the cover slip was exposed to 5 mg/ml streptavidin (Promega) in this space for 15 min, and the remaining free streptavidin was washed away with a solution containing 50 mM HEPES titrated to pH 7.5 and 100 mM NaCl.

An AdiC-containing nanodisc sample was diluted to 30–100 pM, estimated from an evaluation of the absorbance of the sample at 550 nm wavelength against the extinction coefficient of bifunctional rhodamine. The diluted sample was flowed into the space between the assembled cover slip and slide. After allowing a biotin-moiety covalently linked to the N-terminus and the streptavidin-binding tags linked to the N- and C-termini in each of AdiC subunit (i.e., totaling six available attachment points per AdiC dimer) to bind to streptavidin on the cover slip, the space was thoroughly washed to remove unattached AdiC with a solution (pH 5) containing 100 mM NaCl, 100 mM dithiothreitol (DTT, Fisher, BP172), and 50 mM acetic acid titrated to pH 5, without or with arginine at a specific concentration. DDT was used to scavenge oxygen to minimize its adverse impact on the fluorophore's emitting intensity and lifetime.

## Fluorescence polarization microscope and intensity recording

As previously described, the fluorescence polarization microscope was built from a Nikon TIRF microscope (model Ti-E) (*Lewis and Lu, 2019c*). To produce an evanescent field at the surface of the sample coverslip, a 140 mW linearly polarized laser beam (532 nm) generated from a 500 mW laser (Crystalaser CL532-500-S) was directed to pass through a ¼ $\lambda$-plate, which transformed the linear polarization to circular polarization. After passing through a polarization-preserving, high numerical aperture 100× objective (Nikon Achromatic, NA = 1.49), the beam approached the coverslip with an incident angle of 68°, the so-called critical angle that leads to TIR required for the formation of an evanescent field (*Axelrod et al., 1984*). The emission of polarized fluorescent light from individual fluorophores excited by the evanescent field was directed to a 50:50 non-polarizing beam splitter (Thorlabs CM1-BS013) after passing through the objective (*Figure 2*), and then to a 540/593 nm bandpass filter (Semrock FF01-593/40-25) that prevents the propagation of excitation light. One resulting beam was further split by a glass (N-SF1) polarizing beam splitter (Thorlabs CM1 PBS251) along 0° and 90° and the other by a wire-grid polarizing beam splitter (Thorlabs WP25M-Vis) along 45° and 135°. These four emission intensity components, labeled as $I_0$, $I_{45}$, $I_{90}$, and $I_{135}$, were individually directed onto four designated sectors in the CCD grid of an EMCCD camera (Andor iXon Ultra 897), where the four intensities from a given fluorophore appeared in the corresponding positions of the four sections.

Here, fluorescence intensities from individual bifunctional rhodamine molecules, each attached to helix 6A in AdiC, were collected with the microscope and captured every 10 ms with an EMCCD camera at the room temperature 22°C. Following extraction of temporal information from the intensities with the changepoint analysis described below, we applied a Gaussian filter (with a corner frequency of 7.5 Hz) to all four intensity traces to reduce high-frequency noise, where the rise time was 22 ms. From these filtered intensities, we calculated angles as described below.

The experiments were performed on five separate occasions. Data collected among these separate collections are statistically comparable and were pooled together, resulting in sufficiently narrow distributions as illustrated in *Figure 5*. The width of the distributions reflects both technical and biological variations. Outlier data were excluded on the following basis. First, while fluorescence intensity is expected to vary among different polarization directions, the total intensity should not exhibit large variations unrelated to protein-conformational changes, such as more than one step bleaching of fluorescence. Second, for a given recording, at least 15 events are required to obtain a 95% confidence level for state identification, so any short traces with less than 15 events were excluded on the assumption that the short and long traces belong to the same distribution. All traces used contain four states and have an average length of ~4 s. There were no traces with an expected resolvable SNR that did not exhibit transitioning events. Third, for event detection and state identification, an SNR greater

than 5 is necessary for the required minimum angle resolution. Thus, any set of intensity traces with this ratio less than 5 were excluded. The sample sizes were estimated on the basis of previous studies (*Lewis and Lu, 2019c*; *Lewis and Lu, 2019a*; *Lewis and Lu, 2019b*) to yield sufficiently small standard errors of mean to obtain accurate estimate of the mean. Practically, the error bars are comparable to the sizes of the symbols of data as illustrated in *Figures 6 and 10A*. The 95% confidence intervals are provided for all determined equilibrium constants in *Table 1*.

## System parameters

Theoretical aspects of the three-channel polarized emission system have been previously described (*Fourkas, 2001*), and extended to and practically implemented with four channels (*Ohmachi et al., 2012*; *Lippert et al., 2017*; *Lewis and Lu, 2019c*). Briefly, the intensity of a given polarized component with angle $\phi$, $I_\phi$, collected by the microscope's objective from the emission of a fluorophore, which is excited by an evenescent field generated by a circularly polarized laser beam under a TIR condition, is dependent on the fluorophore's orientation. As such, $I_\phi$ is defined by the spherical coordinates $\theta$ and $\varphi$ in a manner that

$$I_\psi = \tfrac{1}{4} g_\psi I_{\text{tot}} \left( X_4\left(\delta\right) \left( \sin^2\theta \left( f_\psi X_1\left(\alpha\right) \cos\left(2\left(\varphi - \psi\right)\right) + X_2\left(\alpha\right)\right) - \tfrac{2}{3} X_2\left(\alpha\right)\right) + X_3\left(\alpha\right)\right) \tag{7}$$

where the polarization angle $\psi = 0°$, $45°$, $90°$, or $135°$. The factor $f_\phi$ corrects for systematic reduction in the maximal achievable anisotropy of the light per channel $\psi$. The coefficients $X_1$, $X_2$, and $X_3$ correct for the incomplete collection of photons by a microscope objective with collection half-angle $\alpha$:

$$\begin{aligned} X_1\left(\alpha\right) &= \tfrac{\pi}{12}\left(7 - 3\cos\alpha - 3\cos^2\alpha - \cos^3\alpha\right) \\ X_2\left(\alpha\right) &= \tfrac{\pi}{2}\left(\cos\alpha - \cos^3\alpha\right) \\ X_3\left(\alpha\right) &= \tfrac{2\pi}{3}\left(1 - \cos\alpha\right) \end{aligned} \tag{8}$$

Ideally, when presented with a beam of non-polarized light, the system splits that beam into four of equal intensity. Small deviations from this theoretical equality are corrected by normalizing each intensity of a given channel ($\psi°$) to that of the 90° channel chosen as the reference here:

$$g_\psi = \frac{I_{\text{tot},90}}{I_{\text{tot},\psi}} \tag{9}$$

Besides those three types of system parameters, the coefficient $X_4$ corrects for the fast diffusive motion ('wobble') of the probe relative to the attached protein, which is measured in terms of the half-angle $\delta$ of the wobble cone:

$$X_4\left(\delta\right) = \tfrac{1}{2}\cos\delta\left(1 + \cos\delta\right) \tag{10}$$

The parameter $\delta$ was experimentally estimated to be 22.5° in a separated macroscopic anisotropy study of the proteins as previously described (*Lewis and Lu, 2019c*).

Analytic solutions have been found for θ, φ and $I_{\text{tot}}$ from *Equation 7* (*Lewis and Lu, 2019c*):

$$\varphi = \tfrac{1}{2}\tan^{-1}\left( \frac{\left(I_{45} - I_{135}\right)\left(f_0\cos 2\psi_0 - f_{90}\cos 2\psi_{90}\right) - \left(I_0 - I_{90}\right)\left(f_{45}\cos 2\psi_{45} - f_{135}\cos 2\psi_{135}\right)}{\left(I_0 - I_{90}\right)\left(f_{45}\sin 2\psi_{45} - f_{135}\sin 2\psi_{135}\right) - \left(I_{45} - I_{135}\right)\left(f_0\sin 2\psi_0 - f_{90}\sin 2\psi_{90}\right)}\right) \tag{11}$$

$$I_{\text{tot}} = \tfrac{1}{2}\left( \frac{\left[X_2 + X_1 f_0\cos 2\left(\varphi - \psi_0\right)\right]I_{90} - \left[X_2 + X_1 f_{90}\cos 2\left(\varphi - \psi_{90}\right)\right]I_0}{X_1\left(X_3 - \tfrac{2}{3}X_2 X_4\right)\left(f_0\cos 2\left(\varphi - \psi_0\right) - f_{90}\cos 2\left(\varphi - \psi_{90}\right)\right)} + \frac{\left[X_2 + X_1 f_{45}\cos 2\left(\varphi - \psi_{45}\right)\right]I_{135} - \left[X_2 + X_1 f_{135}\cos 2\left(\varphi - \psi_{135}\right)\right]I_{45}}{X_1\left(X_3 - \tfrac{2}{3}X_2 X_4\right)\left(f_{45}\cos 2\left(\varphi - \psi_{45}\right) - f_{135}\cos 2\left(\varphi - \psi_{135}\right)\right)}\right) \tag{12}$$

$$\theta = \sin^{-1}\left( \sqrt{ \frac{1}{2X_1 X_4 I_{\text{tot}}}\left( \frac{I_0 - I_{90}}{f_0\cos 2\left(\varphi - \psi_0\right) - f_{90}\cos 2\left(\varphi - \psi_{90}\right)} + \frac{I_{45} - I_{135}}{f_{45}\cos 2\left(\varphi - \psi_{45}\right) - f_{135}\cos 2\left(\varphi - \psi_{135}\right)}\right)}\right) \tag{13}$$

The angle $\theta$ was defined from 0° to 90° and $\varphi$ from 0° to 180° (originally calculated as –90° to 90°) with degenerate solutions of $\theta$ or 180°- $\theta$ and $\varphi$ or 180° + $\varphi$. The solutions for individual events of a given molecule, which were expected to be limited within an appropriately defined quarter sphere because of the expected small angle changes of about 10°–40°, were chosen on the basis of minimizing the variance of the distribution of the resulting angles for the events. When $\theta$ is near 90°, the noise of

intensities might prevent a solution of $\theta$, in which case we simply set the $\theta$ values to 90°. We only analyzed data from the particles with no more than a few percent of such data points.

## Calibrations of the camera

Photons hitting the CCD chip of an EMCCD camera result in the release of individual electrons in accordance with the photoelectric effect. These electrons pass through multiple layers, at each of which they have a probability to cause the release of additional electrons, effectively amplifying the original signal with a gain ($G$) (*Lidke et al., 2005*; *Heintzmann, 2016*). The relationship of the $N$ photons released over a given time interval $\Delta t$ and the recorded intensity $I$ is expressed as

$$I \cdot \Delta t = G \cdot N + \text{offset} \tag{14}$$

As an inherent feature of EMCCD cameras, the offset is the intensity recorded by the camera when the shutter is fully closed. These parameters are estimated by analyzing the relationship between photon intensities recorded at multiple laser intensities versus the corresponding SNR. For photon count $N$, the standard deviation $\sigma$ due to shot noise is given by the square root of $N$. In addition, EMCCD cameras add an additional multiplicative noise that effectively scales the $\sigma$ due to shot noise by a factor of $\sqrt{2}$ that relates the SNR to $N$ via the expression

$$SNR = \frac{I}{\sqrt{2}\sigma} = \sqrt{\frac{N}{2}} \tag{15}$$

Substituting *Equation 15* into *Equation 14* yields

$$I \cdot \Delta t = 2G \cdot SNR^2 + \text{offset} \tag{16}$$

On a plot of $I \cdot \Delta t$ versus $SNR^2$, the slope is twice the gain, whereas the y-intercept is the offset. For our system, the value of the gain was found to be 146 and that of the offset was 220.

Furthermore, when the signal from an interested fluorophore is split from the background signal, the combined intensity signal recorded is related to $N$ by the following relation

$$Signal = G \cdot N + G \cdot N_{bak} + \text{offset} \tag{17}$$

rearranged to

$$N = \frac{Signal - \text{offset}}{G} - N_{bak} \tag{18}$$

where $N_{bak}$ is the number of photons underlying the background signal, which needs to be subtracted. The quantum efficiency (QE) of the Andor Ixon EMCCD camera is estimated as ~0.95 in the visible light range, meaning that 95% of photons contacting the CCD chip are detected. Therefore, the actual photon count is calculated as $N_{det} = N / QE$, and the effective SNR is also scaled accordingly.

## Detection of event transitions

The photon flux intensity recorded in a given channel changes when the orientation of the fluorophore is changed. To detect these transitions within the measured polarized intensities, we adopted a version of the changepoint algorithm (*Chen and Gupta, 2001*; *Beausang et al., 2008*). Such a process was based on calculating a log likelihood ratio over a period of time to determine the maximal ratio that identified the point where the perceived change of photon-release rate occurred, that is, the time at which the fluorophore transitions from one orientation to another. This method has previously been applied to analyzing the photon-arrival time captured on a continuous basis with photon-multiplier-based multi-channel recordings (*Beausang et al., 2008*), which was adapted for analyzing photons collected over a fixed time interval with an EMCCD camera (*Lewis and Lu, 2019c*).

When a camera is used as a detector, photons emerging from each channel are effectively binned over each frame. A series of consecutive $k$ frames with a constant exposure time ($\Delta t$) is expressed as

$$t_0, t_1, \ldots, t_i, t_{i+1}, \ldots, t_{m-1}, t_m = 0, \Delta t, \ldots, i\Delta t, (i+1)\Delta t, \ldots, (k-1)\Delta t, k\Delta t \tag{19}$$

For a given frame $i$, the intensity $I_i$ is defined by the rate of photon release:

$$I_i = \frac{n_i}{\Delta t} \tag{20}$$

where $n_i$ is the number of photons release within frame $i$. The cumulative distribution, $m_j$, is built by adding the number of photons for the successive time frames:

$$m_j = \sum_{i=0}^{j} n_i \tag{21}$$

If during an interval $T$ the rate changes from $l_1$ to $l_2$ at the time point $\tau = i\Delta t$, and the number of emitted photons prior to this change is $m$ (**Equation 21**) and $N - m$ after the change, then the likelihood ratio of a transition occurring at frame $i$ in the log form is described by

$$LL_R = \sum_{r=1}^{h} \left( m_r \ln\left(\frac{m_r}{\tau}\right) + \left(N_r - m_r\right) \ln\left(\frac{N_r - m_r}{T - \tau}\right) - N_r \ln\left(\frac{N_r}{T}\right) \right) \tag{22}$$

where $h$ equals 4, the number of emission channels in the system. We set the threshold of significance for $LL_R$ at the level that limits the false-positive events to 5% on the basis of simulation studies for the corresponding levels of SNR, and the resulting false-negative events were about 1%.

The program was started by identifying one transition over the entire trace. If a change-point X was identified, it would then search for additional transitions between the start of the trace and point X and between X and the end. This iterative search with successively shortened stretches continued until no more transitions were identified.

## State identification

Following the detection of intensity transition time points using the changepoint method and subsequent calculation of angles, the states of individual events, each demarcated by two consecutive transition timepoints, were identified on the basis of x,y,z values, calculated from the corresponding $\theta$ and $\varphi$ values, along with $r$ of a unity value, in accordance with the transformation relations between the Cartesian and spherical coordinate systems:

$$x = r \sin\theta \cos\varphi$$
$$y = r \sin\theta \sin\varphi \tag{23}$$
$$z = r \cos\theta$$

Given the $r$ of a unity value, which carries no information regarding spatial orientation, for all cases, x,y,z in all cases would always be on a unit sphere and fully encode the orientation information specified by $\theta$ and $\varphi$.

The identification of the states is done by using a 'nearest-neighbor' method in a Cartesian coordinate system, where a given event was assigned to the closest state distribution. Closeness was determined by the minimum distance $d_{i,k}$ between the x,y,z coordinates of the $i$th event and the distribution means $\langle x_k \rangle$, $\langle y_k \rangle$ and $\langle z_k \rangle$, where $k$ is the number of potential states, calculated as:

$$d_{i,k} = \sqrt{\left(\langle x_k \rangle - x_i\right)^2 + \left(\langle y_k \rangle - y_i\right)^2 + \left(\langle z_k \rangle - z_i\right)^2} \tag{24}$$

Minimization was performed by using a k-means clustering algorithm optimized with two coupled algorithms (*simulated annealing* plus *Nelder–Mead downhill simplex*)(**Press et al., 2007**), as we described previously in more detail (**Lewis and Lu, 2019c**). The resulting four conformational state distributions are denoted as $C_1 - C_4$.

States must be indexed so that they are always numerated in the same sequential order among individual molecules, a prerequisite for correctly relating the states identified here to those identified crystallographically. This indexing is based on the shortest, if not straightest, path distance between the first and last states, $C_1$ and $C_M$ (where $M = 4$ in our case) as they detour through the remaining states (**Figure 7—figure supplement 1B**). This path length is calculated as the sum of the cumulative distances between two adjacent states, denoted as $d_{tot}$:

$$d_{tot} = \sum_{i=1}^{M-1} d_{i,i+1} \tag{25}$$

where the number 1 indicates a chosen starting state and the distance $d_{i,j}$ between states indicated by positions $i$ and $j$ in a Cartesian coordinate system is defined as:

$$d_{i,j} = \sqrt{\left(x_i - x_j\right)^2 + \left(y_i - y_j\right)^2 + \left(z_i - z_j\right)^2} \tag{26}$$

If the states were located along a perfect line, the total path distance would equal the distance between states 1 and M, that is, $d_{1,M}$, where $d_{1,M} = d_{tot}$. However, if they were not on a line, $d_{tot}$ would be greater than $d_{1,M}$ by $\Delta d$ such that

$$\left(\sum_{i=1}^{M} d_{i,i+1}\right) - d_{1,M} = \Delta d \tag{27}$$

Upon calculating $\Delta d$ for each of the 24 possible sequences relating the four states, the shortest path can be found on the basis of the smallest value of $\Delta d$, which has two solutions with equal path length, that is, 1-2-3-4 versus 4-3-2-1 (*Figure 7—figure supplement 1C* vs. *Figure 7—figure supplement 1D*). The sequence of $C_1$-$C_2$-$C_3$-$C_4$ shown in *Figure 7—figure supplement 1C* was consistently chosen.

### Transformation of the laboratory framework to a local framework

As explained in the text, individual molecules did not have the same orientation. Thus, a direct comparison among them requires all molecules be rotated from the laboratory frame of reference into a common local frame of reference (*Figure 7—figure supplement 1A*). In the local frame, the x,y-plane is defined by the vectors representing $C_1$ (**V₁**) and $C_4$ (**V₄**) and the x-axis is defined by **V₁**. The z-axis is perpendicular to the x,y-plane. The **x**, **y**, and **z**-axes, represented by unit vectors, are defined by:

$$X = \frac{V_1}{|V_1|}; \quad Y = \frac{X \times V_{N_{states}} \times X}{|X \times V_{N_{states}} \times X|}; \quad Z = \frac{X \times Y}{|X \times Y|} \tag{28}$$

where

$$V_i = \begin{pmatrix} x_i \\ y_i \\ z_i \end{pmatrix} \tag{29}$$

$\theta$ and $\varphi$ in the local frame of reference are then calculated as:

$$\theta_i = \cos^{-1}\left(\frac{\mathbf{Z}}{|\mathbf{Z}|} \cdot \frac{\mathbf{V}_i}{|\mathbf{V}_i|}\right) \tag{30}$$

$$\varphi_i = \sin^{-1}\left(\frac{\mathbf{Y}}{|\mathbf{Y}|} \cdot \frac{\mathbf{V}_i^{proj}}{|\mathbf{V}_i^{proj}|}\right) \tag{31}$$

where

$$\mathbf{V}_i^{proj} = \frac{\mathbf{Z} \times \mathbf{V}_i \times \mathbf{Z}}{|\mathbf{Z} \times \mathbf{V}_i \times \mathbf{Z}|} . \tag{32}$$

### Calculation of the direct angle change Ω between two states

The direct angle change $\Omega_{i,j}$ between two states $i$ and $j$, as represented by the vectors **V**$_i$ and **V**$_j$ defined above, can be calculated from the relation:

$$\Omega_{i,j} = \cos^{-1}\left(\frac{\mathbf{V}_i \cdot \mathbf{V}_j}{|\mathbf{V}_i| \cdot |\mathbf{V}_j|}\right) \tag{33}$$

## Acknowledgements

This study was supported by the grant DK125521 from the National Institute of Diabetes and Digestive and Kidney Diseases.

## Additional information

### Funding

| Funder | Grant reference number | Author |
| --- | --- | --- |
| National Institute of Diabetes and Digestive and Kidney Diseases | DK125521 | Zhe Lu |

The funders had no role in study design, data collection and interpretation, or the decision to submit the work for publication.

### Author contributions

Yufeng Zhou, Data curation, Formal analysis, Validation, Investigation, Methodology, Writing – review and editing; John H Lewis, Data curation, Software, Formal analysis, Validation, Investigation, Methodology, Writing – review and editing; Zhe Lu, Conceptualization, Supervision, Funding acquisition, Investigation, Writing - original draft, Project administration, Writing – review and editing

### Author ORCIDs

Zhe Lu (iD) http://orcid.org/0000-0001-7108-9303

### Decision letter and Author response

Decision letter https://doi.org/10.7554/eLife.82175.sa1
Author response https://doi.org/10.7554/eLife.82175.sa2

## Additional files

### Supplementary files

• MDAR checklist

### Data availability

Source-data files for all relevant figures are provided.

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
