## [Editor Report]

This study uses a single-molecule polarization microscopy approach to identify the different conformation states that the arginine/agmatine antiporter AdiC transitions through during transport. Four states are identified and proposed to correspond to the key steps in the transport cycle, including inward-open, inward occluded, outward occluded and outward open, setting the stage for measurements of equilibrium constants and kinetics associated with transport. This is a cutting-edge and challenging approach that offers the potential for obtaining direct information of protein conformational equilibria that will be of interest to anyone studying membrane transport mechanisms.

---

## [Decision Letter]

**Decision letter after peer review:**

[Editors’ note: the authors submitted for reconsideration following the decision after peer review. What follows is the decision letter after the first round of review.]

Thank you for submitting the paper "Tracking multiple conformations occurring on Å-and-millisecond scales in single amino-acid-transporter molecules" for consideration by *eLife*. Your article has been reviewed by 4 peer reviewers, including Janice L Robertson as Reviewing Editor and Reviewer #1, and the evaluation has been overseen by a Senior Editor.

Comments to the Authors:

We are sorry to say that, after consultation with the reviewers, we have decided that this work will not be considered further for publication by *eLife*.

The reviewers agreed that your study presents a cutting-edge approach that offers the potential for monitoring conformational equilibria of proteins, such as membrane transporters. In particular, the observation of arginine linked changes in the polarization distributions of AdiC is promising. However, major questions were raised during the review that indicate additional investigation is necessary to test whether the single-molecule data, and post-processing, leads to a robust determination of conformational states that pertain to transport. All of the points are listed in the comprehensive reviews below, but the major concerns are summarized here:

1. In both studies, it is reported that Ex-Ix, i.e. external-occluded to internal-occluded or C1-C4, transitions occur in the apo condition. The analysis also shows that equilibrium between these states (K4,1) is comparable to that in substrate saturating conditions. Therefore, this presents a model for AdiC that is an uncoupled uniporter. This means that if arginine is present on one side of the membrane, without any substrate on the other side, then AdiC would be expected to dissipate the arginine gradient because it could return back to the substrate-loaded side via the apo transition. However, Fang et al., JBC, 2007, carried out these experiments and showed that AdiC does not transport unless there is substrate on both sides of the membrane. Therefore, AdiC is a coupled antiporter that does not transition between the Ex-Ix states without substrate. Since the current study reports the observation of apo occluded state transitions, it raises a major concern that the states that have been identified do not represent the states in the full transport cycle. Further investigation is needed to consolidate the proposed model with the strictly coupled mechanism that has already been demonstrated for AdiC.

2. The conversion of the single-molecule trajectories to state transitions requires substantial post-processing, including averaging, step detection and clustering. Numerous reviewers raised the question of whether the four-state model that is being proposed is objectively determined from the traces or is imposed by restraining the k-means clustering analysis. Along these lines, testing alternate state models and comparing other clustering algorithms that do not require a pre-classification of states is necessary here in order to know if the 4-state model is robust, or equally likely compared to other multi-state models.

3. Following this, the study would be greatly strengthened if additional validation was carried out to test the identification of states, and perturb AdiC transport in a known way. Can a certain state be cross-linked to see whether the single-molecule trajectories report on the isolation of a single state? For perturbing transport, there are examples of mutations or conditions (i.e. high pH, different substrates) that are known to slow transport. Is the data capable of reporting on these changes, and do they agree with the expected shift in transport behavior? These validation studies are important to test the robustness of the single-molecule data and the state models to inform on the overall transport cycle.

4. Finally, macroscopic transport assays of AdiC would be extremely informative alongside the single-molecule studies. This would allow for important controls that will test transport activity of the actual construct being studied here, as well as help to consolidate any discrepancies in the protein behavior that was raised above.

*Reviewer #1 (Recommendations for the authors):*

1. Confirmation of function after labeling. The version of AdiC that is being studied here is labelled by a bis-maleimide-TMR, solubilized in POPC nanodiscs, removed 2 native cysteines, N-terminal Avi-tag and Strep-tag, and C-terminal Strep-tag. However, I do not see that a validation has been carried out to test if this construct/conditions independently yield the same KD for arginine or agmatine binding, or similar transport properties. Experiments of binding (e.g. by ITC) and transport (e.g. by radioactive uptake assays in proteoliposomes) of the labelled construct that is currently being studied are important controls to validate whether the single-molecule polarization data can be compared to wild-type data published previously.

2. The effect of anchoring AdiC on the slide. The strategy for conjugating the sample to the slide is via Strepavidin binding to the protein termini directly. The need for anchoring the protein to slide in order to observe polarization changes is understood. However, it also raises questions whether this anchoring may bias the observation of conformational equilibria. Since there is a small linker at both N- and C-termini, some benchmarking studies would be appropriate to test whether the linker length impacts the equilibria. Shortening the linkers may shift the balance to stabilize certain states, and lengthening (to a limit, within the range where polarization changes can still be observed) would demonstrate if more states become observable and the number of states becomes invariant with linker length.

3. Background labeling? Considering the labeling conditions of pH 8.0 and 4 hours, there is a possibility of background labeling of the maleimides with primary amines. Was the background labeling measured, for instance with constructs that do not contain the reactive cysteines? Also, it was mentioned that bleaching steps were used to assess labeling, but this was not shown so it is unclear what information was being obtained from these analyses. Both of these points should be addressed to give confidence that the label is attaching in the mode described and that there are no other possibilities of non-specific labeling.

4. Robustness of the clustering results. The clustering, as indicated in Figure 5, is not evident by eye and appears more as a continuum of conformations. It also reasons that the clustering results are dependent on the averaging, changepoint detection and k-means clustering parameters. Thus, the robustness of the identification of states should be shown across a range of parameters for each analysis step. In addition, testing different state models (e.g. 2-8 states) would be important to understand if the 4-state clustering is robustly determined or other state models also suitably fit the data.

5. Validation of the model with mutations that are known to shift the equilibrium between states. While the results showing the linkage of the states to arginine binding corresponding to the macroscopic measurements of AdiC KD provides some supportive evidence, additional investigation that tests the different states would strengthen the proposed model. For example, can a mutation be engineered, or reagent applied that stabilizes the inward-open or outward-open state, to see that conferred in the single-molecule polarization traces?

6. A cartoon of the full gene sequence and schematic of the AdiC construct that is being used here would be useful as supplementary information, in order to understand the sizes of tags and linkers applied to the protein.

7. In Figure 9, the colors in the plot do not appear to match the legend (different shades of blue).

8. Since the data is analyzed over single trajectories, how long are these traces typically? How long do the traces have to be in order to observe enough transitions between states?

9. The fitting of the KDs for states C1 and C4 are not justified as there does not appear to be a significant change in the probabilities over the arginine titration. Therefore, they should not be reported with confidence. Along these lines, are the errors that are being reported in Table 1 on the best-fits or are they over different samples?

*Reviewer #2 (Recommendations for the authors):*

Moreover, the identified 4 states and their substrate affinities reported in this manuscript do not significantly advance our understanding of Adic and the APC superfamily. Therefore, I would suggest combining this and the companion manuscript into a single paper. In addition, some more specific questions and suggestions are:

1) The key difference of the data processing described in this manuscript compared to the earlier published work is the introduction of the normalization step. Due to the random X-Y orientation of the transporter proteins on the surface, each protein molecule needs to be analyzed individually to get its own theta and phi angles in the laboratory frame (Cartesian coordinate). Normalization of the angles based on one of the identified states allows to directly compare the results for different particles directly. This is an interesting improvement, so the authors may need to mention this in the abstract.

2) In principle, the normalization may improve the accuracy of the mean angle values even as the deviations might increase. It would be helpful if the authors compared the ensemble-averaged mean values and the single-trace mean values.

3) From my understanding, to better differentiate the states, traces visiting all 4 states should be analyzed. When the authors select the traces with at least 12 transition events, do the events include transitions between all four states? The authors might want to clarify.

4) It is unclear how the authors establish the number of distinct states to be considered in the system. Is the number of states based on the expected structurally resolved 4 states or obtained from the analysis of the traces themselves? There is a proposed so-called full occluded state. Could/should this state be included in the data analysis?

5) The fluorophore is attached via two cysteines. Therefore, in the same conformation, the fluorophore has two orientations related by 180-degree rotation. Will this affect the detected signal? If not, then would a protein conformational change, which makes the fluorophore rotate by 180 degrees, be detected?

6) The protein is modified by a bi-functional fluorophore and surface-immobilized with multiple binding tags. How do you evaluate whether these manipulations affect the state populations and transport activity? The structures of Adic were all solved in the outward-facing states. Simulation (PLoS ONE 11(8): e0160219) also indicated that the inward-facing states, especially the inward-open state, are relatively unstable, suggesting that the population of the inward-facing states should be low. However, in Figure 7B, the inward-open state is highly populated, which seems inconsistent with the expectations. It seems that some additional experiments and or more extensive discussion are needed to support the authors' estimates of the state populations.

7) The state assignments are based on the similarity of the theta and phi angles to those calculated from the four crystal structures. To me, this does not seem entirely convincing, especially regarding the inward-facing states, expected to be lowly populated. The authors might need to use mutants or ligands to stabilize the inward-facing states to verify the assignment.

8) The authors need to describe how they estimate the errors in Table 1.

*Reviewer #3 (Recommendations for the authors):*

1. This work monitors only a single transmembrane helix (helix 6a) in AdiC. This was presented as an advantage, but in reality it also creates strong limitations for data interpretation. By comparing the orientations of helix 6a in PM-identified states (C1-C4) and in structural states, C1-C4 were dubbed externally open (Eo), externally occluded (Ex), internally open (Io), and internally occluded (Ix) states. Such assignment comes with high uncertainty even in optimal conditions. For example, without "seeing" the rest of the protein, how do the authors know that a particular PM state is not an intermediate state not observed in structural studies? Such a concern arises in part because of the assignment of C1 as Ex and C4 as Ix, with a state possibility of 0.19 and 0.14, respectively. Why would an antiporter sample occluded states so often without ligands? Is it possible that C1/C2 are both externally open (with some differences in helix 6a orientations) and C3/C4 both internally open and the occluded states are too rare to be observed when there is no ligand?

2. The assignment of PM states was actually done under not-so-optimal conditions. AdiC structures were only determined in ligand-free Eo and ligand-bound Ex states, so the authors used BasC and ApcT structures to represent the Io and Ic states of AdiC. Multiple questions about data treatments follow naturally. What's the justification to compare ligand-free PM states with ligand-bound structural states (Figure 6)? Although BasC (complexed with a nanobody) and ApcT have similar LeuT fold like AdiC, their sequences are not particularly similar and it's conceivable that AdiC's helix 6a might have significantly different orientations in Io and Ic states than the helix 6a in BasC and ApcT. How would such a possibility affect the assignment of the PM states?

3. Another main issue is about the relation of ligand-free and ligand-bound PM states. There are four ligand-bound PM states, labeled as C1L, C2L, C3L, and C4L to indicate that they correspond to the ligand-free C1, C2, C3, C4 states. However, the ligand-free and ligan-bound PM states appear to have quite different *θ*/*ψ* profiles (Figure 5), and the latter were not compared to the structures. How did the authors link ligand-bound PM states to the structural states? How to know for sure that C1 is related to C1L, C2 to C2L, etc? Moreover, it could be misleading to distinguish the corresponding ligand-bound and -free states just by an "L" as if their only difference is the ligand – the technique here provides no such resolution.

4. The authors might want to consider utilizing additional strategies to verify their state assignments. One possibility is to use structures as guidance to lock AdiC in a particular state via Lysine crosslinking and see how such maneuver affects PM results. The determined KD reflects how ligands drive the protein from ligand-free to ligand-bound conformational equilibria without providing much insight into the conformational details in each set of equilibrium.

*Reviewer #4 (Recommendations for the authors):*

In this paper, Zhou et al. propose a polarization microscope for measuring the emission polarization of bifunctional rhodamine molecules attached to AdiC transporters. The polarization is used to resolve the orientation of the fluorophores, which allows the authors to successfully resolve the four conformations of AdiC at a temporal resolution of tens of milliseconds. The measured orientation for each conformation is validated with the results using crystallography.

Overall, I believe the paper is well written and demonstrates a great application for orientation imaging using polarized microscopes. Detailed experimental procedures, calibrations, and mathematical frameworks are included. I have the following recommendations to improve the manuscript.

1) On page 20, the authors note that they set a threshold to filter out molecules whose total intensity varies during the measurements. The statement that "while fluorescence intensity is expected to vary among different polarization directions, the total intensity should be essentially invariant" is not true. Since the authors use TIRF illumination to excite the molecules, the excitation polarization component along the tilting direction (e.g., along the y-axis) of the excitation is 0, i.e., molecules oriented along that direction (e.g., y-oriented) will be excited less effectively compared to other orientations.

2) Could the authors provide more details regarding how the clusters are ranked? The authors note that C1-C4 are "ranked according to the values of both angles". It is not clear to me how this is done. Also, what is the range of the measured theta_L and phi_L? And how is the warping of the spherical coordinates handled in the ranking process, e.g., a change from 350 deg to 10 deg is +20 deg or -340 deg.

3) Is the k-means clustering also based on the distance in the Cartesian space, similar to the state identification?

Following comment (1) from above, could the authors comment on the possibility of further improving the measurement precision and accuracy using the excitation-dependent total intensity? Since the authors report a wobble angle of 22.5 degrees, the excitation dipole moments should be mostly aligned with the emission dipole moments.

---

## [Author Response]

[Editors’ note: The authors appealed the original decision. What follows is the authors’ response to the first round of review.]

1. In both studies, it is reported that Ex-Ix, i.e. external-occluded to internal-occluded or C1-C4, transitions occur in the apo condition. The analysis also shows that equilibrium between these states is comparable to that in substrate saturating conditions. Therefore, this presents a model for AdiC that is an uncoupled uniporter. This means that if arginine is present on one side of the membrane, without any substrate on the other side, then AdiC would be expected to dissipate the arginine gradient because it could return back to the substrate-loaded side via the apo transition. However, Fang et al., JBC, 2007, carried out these experiments and showed that AdiC does not transport unless there is substrate on both sides of the membrane. Therefore, AdiC is a coupled antiporter that does not transition between the Ex-Ix states without substrate. Since the current study reports the observation of apo occluded state transitions, it raises a major concern that the states that have been identified do not represent the states in the full transport cycle. Further investigation is needed to consolidate the proposed model with the strictly coupled mechanism that has already been demonstrated for AdiC.

In this manuscript, as outlined above, we mainly demonstrate the application of the method in a membrane protein. Additionally, we evaluated the energetic differences among the resolved states in the form of equilibrium constants calculated from state probabilities. This operation does not require the kinetic information regarding state connectivity. As such, we now place a dash line (without any arrow) between a given pair of states to solely indicate an equilibrium constant for a pair of neighboring states (Figure. 10B). The connectivity was quantitatively dealt with in the original companion paper. Given that we limited the scope here, this comment would no longer be directly applicable here but is applicable to the kinetic study that we will revise for a later, separate publication.

Regarding the hypothesis that AdiC is an obligatory exchanger, we will discuss, in a transparent and analytic manner, the experimental evidence that disapproves this hypothesis when we address, in the future, the kinetic mechanism underlying AdiC’s conformational changes and its transport of substrate in the revised version of the original companion manuscript.

2. The conversion of the single-molecule trajectories to state transitions requires substantial post-processing, including averaging, step detection and clustering. Numerous reviewers raised the question of whether the four-state model that is being proposed is objectively determined from the traces or is imposed by restraining the k-means clustering analysis. Along these lines, testing alternate state models and comparing other clustering algorithms that do not require a pre-classification of states is necessary here in order to know if the 4-state model is robust, or equally likely compared to other multi-state models.

Our inadequate presentation apparently made the reviewers think that we might have only evaluated the conformation model with four (macroscopic) states. In fact, as described previously in the cited paper (Lewis and Lu, 2019c, NSMB), we evaluated all possible numbers of state that can be resolved with the common microscopy resolution criterion defined as a separation of 2.5σ (standard deviation) between the peaks of two state populations (‘Resutls’), here in terms of spatial orientation (*θ* and *ϕ* ). The analysis is done without any preconceived kinetic models (now stated in ‘Results’), including the common four-state transport model.

As stated in ‘Results’, we first detected the fluorophore orientation change with respect to two stationary polarizers from the relative changes in photon counts that occur simultaneously in all four different, resulting polarization components using the changepoint algorithm. Unlike other types of fluorescence-based method, such as FRET, the requirement of concurrent changes in all four channels, here, markedly increases the confidence of the detection of transitions. Two consecutive transition points demarcate an event during which the protein molecule adopts a specific conformation. We then calculated the mean *θ* and *ϕ* angles of a given event from all four intensities. This calculation is done with ratios of intensities, so the same result is achieved whether we use the total number or the average number of photons recorded from each channel during that event (‘Results’). It is noteworthy that unlike random noise, the directions of intensity changes between a pair of intensity channels should be related in a specific way, which is another advantageous feature, compared to other types of fluorescence-based method. For example, when the fluorophore’s mean electrical-field vector moved such that its *ϕ* angle increased in the direction of 0 to 90°, the intensity recorded in the 0° channel would decrease whereas that in 90° should increase accordingly; the intensities of 45° and 135° channels would also characteristically move in opposing directions (‘Results’).

After the angles of individual events were calculated, we separated them into different populations on the basis of the information of *θ* and *ϕ* angles together in a three-dimensional operation, as mentioned above and detailed in ‘Materials and methods’; we now explain the essence of the three-dimensional sorting in ’Results’ as well. It is practically impossible to identify the state from either a *θ* trace or a *ϕ* trace alone. Thus, we used the *k*-means clustering (i.e., shortest distance) algorithm to assist us to perform this 3D-sorting task in a Cartesian coordinate system, which would be extremely challenging for us to do manually on subjective visual guidance, if possible. Our program examined the case of one-state distribution to the case of *k*-state distributions, one at a time in a series of separate operations. These operations were done on the basis of the information of both *θ* and *ϕ* together so as to reach the maximal resolvability for the present study and high confidence. For each successful operation, the resolution criterion was ensured, i.e., peaks of all populations were separated by at least 2.5σ. By this common criterion, the highest number of resolvable states was four at each examined substrate concentration, so the minimal conformation model here must have four states for either apo or substrate-bound forms (shown now in Figures 5 and 6), totaling eight states. Again, we did not pick four states because of the knowledge that a typical transporter is expected to have four types of conformation.

Given that we tracked all four states, we could determine their probabilities in all examined concentrations of Arg^+^. From these probabilities, we can in turn determine all the equilibrium constants that quantitatively and fully defines the four-state energetic model of conformational changes. Furthermore, the probabilities of conformations *C*_2_ and *C*_3_ clearly varied with the Arg concentration whereas the other two states varied little, *C*_2_ and *C*_3_ should be two open states that could directly interact with substrates, and the remaining two should be the occluded states (in adopting the conventional name)(‘Discussion’), four of which are arranged in a symmetric manner. We do not need any other information to reach this point. Note that the relative invariability of the probability of *C*_1_ or *C*_4_ with the substrate concentration does not mean these two states do not bind substrates. In the present case of multistates, this invariability actually indicates that they do bind substrates because if *C*_1_ or *C*_4_ did not bind Arg*^+^* with any meaningful affinity while other states do, *C*_1_ or *C*_4_ would practically vanish in a saturating concentration of Arg*^+^*.

Regarding the sidedness of the model (‘Discussion’), the relation between the orientations determined from polarization measurements and those from the structures of known states helps distinguish between the so-called internal and external conformations. The external open and external occluded structures of AdiC are available. Fortunately, for these two states, the two methods yield consistent orientations. Furthermore, we also used the internal open and internal occluded structures of two related transporter as proxies, or space holders, for the corresponding states of AdiC in the exercises to illustrate how to perform multistate spatial alignment.

3. Following this, the study would be greatly strengthened if additional validation was carried out to test the identification of states, and perturb AdiC transport in a known way. Can a certain state be cross-linked to see whether the single-molecule trajectories report on the isolation of a single state? For perturbing transport, there are examples of mutations or conditions (i.e. high pH, different substrates) that are known to slow transport. Is the data capable of reporting on these changes, and do they agree with the expected shift in transport behavior? These validation studies are important to test the robustness of the single-molecule data and the state models to inform on the overall transport cycle.

In our first study of AdiC, we chose to perturb the system in the arguably most naturally consequential way, i.e., by examining AdiC in various concentrations of its natural substrate Arg^+^, totaling 13 concentrations (Figure 10A). From the concentration dependence, we determined the conformations that substrates could directly access versus those that they could not directly access, and their energetic characteristics. The examination of the impact of pH and mutations on AdiC’s properties is the natural progression of the present study, and will undoubtedly be informative. Each type of this examination is a major undertaking. As such, we plan to perform these secondary studies in a manageable manner.

If the molecules were “frozen” in one conformation, we could no longer align individual molecules in the absence of other conformations as references. At a minimum, individual molecules and thus the attached probes are expected to be randomly orientated on the sample-supporting glass in term of the rotation angle *ϕ* around the z-axis, even without considering the variation in the inclination angle *θ* yet. We thus would not be able to obtain a mean spatial orientation among all particles for comparison with the expected orientation for the conformation. Furthermore, each of the four channels would have a statistically constant intensity. Then, we would not be able to tell whether a protein molecule were in a frozen state versus in a defunct state caused by the chemical reaction. As outlined above, we do not need to freeze the molecule in a state to identify the types of tracked conformational states, which can be tested in future.

The outcomes of our study are comparable to published estimates of K_d_ (in this manuscript) K_m_ and “k_cat_” (in the original companion manuscript) in that: our estimates are within the range of K_d_ measured with ITC (‘Discussion’), which is a factor of 3 for the Miller lab alone; our estimated K_m_ value on average is 2.38 fold smaller than the average of functional estimates; our estimated “k_cat_” on average is 1.25 fold less than the average of functional estimates. These ratios would correspond to 0.87 kT and 0.22 kT if they were evaluated as errors between the two approaches, in terms of the underlying substrate binding energy and the activation energy, respectively. For reference, according to the equipartition theorem, the average energy of individual atoms in a monoatomic ideal gas in thermal equilibrium is ~1.5 kT. Arguably, the system and statistical errors of their flux assay in terms of K_m_ should not be less than the observed error range of ITC that is a factor of 3 for the Miller lab. The upper limit of this range of K_d_ differs from that of another lab by a factor of >2. If so, the differences between our estimates and the function-and-biochemical estimates of the Miller lab are within the system and statistical errors of their approaches. As such, our approach has independently led to comparable estimates, and our estimates are thus valid.

4. Finally, macroscopic transport assays of AdiC would be extremely informative alongside the single-molecule studies. This would allow for important controls that will test transport activity of the actual construct being studied here, as well as help to consolidate any discrepancies in the protein behavior that was raised above.

As described below, our method can independently and straightforward estimate the K_d_ K_m_ and “k_cat_”, and as described above, our estimates are comparable to existing estimates. We do not know any discrepancies between our estimates and existing estimates to the levels that concern us. We mutated two native cysteine residues because removing all native cysteine residues have no major impact on the functional properties (Tsai et al., Biochemistry 2012). A bi-functional rhodamine molecule was attached via two mutant cysteine residues to helix 6a in the region extracellular to the substrate-binding site to avoid affecting the binding affinity. Furthermore, as we explained in ‘Results’, spacer sequences were put between tags and between a tag and the AdiC sequence to introduce flexibility to the extent that the inclination angle varied considerably among molecules. Now stated in ‘Results’, as assessed with ITC, the protein resulting from the cDNA construct genetically engineered for the present purpose exhibited a K_D_ of 104 μM for Arg^+^ (Figure 1—figure supplement 1D), which is within the previously reported range of 32 to 204 μM for AdiC.

With more than a decade of effort, we developed the present high-resolution fluorescence-polarization method to track the conformational dynamics of a protein. Structural conformations of a protein are the physical basis of its functional properties. If so, accurate conformational information of a protein acquired with adequate spatial and temporal resolutions is expected to report the corresponding functional behaviors. This expectation should be fulfillable without the guidance of any concurrent assays of functional properties at any steps, before the final outcomes are yielded for comparison with independently acquired functional data. Upon the completion of development of our method, we test if we could fulfill this expectation for a successfully developed method, within the expected errors of the data independently obtained by others with reference methods. Also, such a test would be a typical validation approach in physics, i.e., carrying out the entire process completely independent of the reference method.

As a first test, we chose to examine the isolated gating ring of the Ca^2+^-dependent MthK channel (Lewis and Lu, 2019A-C, NSMB), because the electrophysiological approach tends to yield more accurate measurements than many other types of functional measurements, such as flux assays. Based on polarization measures, without any concurrent examination of functional properties of the channel, i.e., ionic current, during data collection and deduction, we have successfully established an analytic model that quantitatively predicts the P_o_-[Ca^2+^], k_on_-[Ca^2+^], and k_off_-[Ca^2+^] of relations of the whole channel, within the standard errors of individual data points previously published by others. In a mean-to-mean comparison, the biggest discrepancy in terms of free-energy difference is only about ¼ kT, which occurred at an electrophyisiological data point measured under a condition that is expected to yield less accurate data. This comparison shows a high degree of accuracy of our method in terms of determining state probabilities, lifetimes, and connectivity, from which equilibrium constants and rate constants are calculated.

Here, as a second test, we try to test our method on a membrane protein using the AdiC transporter. The method again allows very accurate determination of, e.g., state probabilities (with standard errors being about or less than the size of the data symbols; Figure. 10A), and thus that of the equilibrium constants (*K*) for individual states. For reference, the error range of ΔG, estimated from kTlnK+σK−σ or kTlnK+2σK−2σ, is 0.16 or 0.32 kT. Furthermore, in the original companion manuscript, we determined the rate-limiting first-order transition. From the difference between the forward and backward rates of this transition alone, we can calculate the maximum net flux rate of a given substrate species (“k_cat_”). These two forward and backward rates are solely determined by the two underlying rate constants and probabilities of the two connected states, which we have determined accurately, judging from standard errors. From the substrate-concentration dependence of the net flux rate, we can estimate K_m_. Thus, our method enables us to independently estimate K_d_, K_m_ and k_cat_ values in a straightforward manner, and to compare the final outcomes with those previously determined by the Miller lab using ITC and flux assays. The system and statistical errors of flux assay are generally large, much greater than those of electrophysiological recordings. Nonetheless, we are pleased to learn that the differences between our estimates and those estimated using ITC and the flux assay by the Miller lab are within the expected error range of their assays, as mentioned above. Thus, all of the technical maneuvers, which we made, had apparently limited impacts on the protein conformation-energetic and -kinetic properties, maneuvers including mutations and the attachment of protein molecules to a glass surface via tags, which are part of what may contribute to the system errors of our approach. Again, if the mutations or tags had huge impacts on the conformational energetic and kinetics, we would see much greater differences in the comparison of our estimates with their estimates. The fact that our model is solely reached from and fully constrained by polarization measurements alone gives us, and hopefully others as well, the confidence that the resulting relation between our estimates by the polarization method and those of the Miller lab by a functional method objectively reflects the total system-and-statistical errors of the two compared approaches, which are within the expected error range of their approach alone.

Reviewer #1 (Recommendations for the authors):1. Confirmation of function after labeling. The version of AdiC that is being studied here is labelled by a bis-maleimide-TMR, solubilized in POPC nanodiscs, removed 2 native cysteines, N-terminal Avi-tag and Strep-tag, and C-terminal Strep-tag. However, I do not see that a validation has been carried out to test if this construct/conditions independently yield the same KD for arginine or agmatine binding, or similar transport properties. Experiments of binding (e.g. by ITC) and transport (e.g. by radioactive uptake assays in proteoliposomes) of the labelled construct that is currently being studied are important controls to validate whether the single-molecule polarization data can be compared to wild-type data published previously.

These comments have been addressed above under the responses to summative comment #4.

2. The effect of anchoring AdiC on the slide. The strategy for conjugating the sample to the slide is via Strepavidin binding to the protein termini directly. The need for anchoring the protein to slide in order to observe polarization changes is understood. However, it also raises questions whether this anchoring may bias the observation of conformational equilibria. Since there is a small linker at both N- and C-termini, some benchmarking studies would be appropriate to test whether the linker length impacts the equilibria. Shortening the linkers may shift the balance to stabilize certain states, and lengthening (to a limit, within the range where polarization changes can still be observed) would demonstrate if more states become observable and the number of states becomes invariant with linker length.

As stated at the beginning of our responses and now more clearly in ‘Results’ of the manuscript, we deliberately introduced flexible spaces such that the molecules are not strained and adopted differing orientations. The attachments did not appear to be markedly impactful because our estimated K_d_ values by ITC or by the polarization method are comparable to preciously reported ITC values. (Furthermore, the kinetic parameters yielded from the simulated fluxes using the rate constants and probabilities are within the expected error range, as reported in the original companion manuscript. These simulations were done under the conditions that the concentrations on both sides of the membrane were recalculated for each time point, i.e., they were not fixed.)

3. Background labeling? Considering the labeling conditions of pH 8.0 and 4 hours, there is a possibility of background labeling of the maleimides with primary amines. Was the background labeling measured, for instance with constructs that do not contain the reactive cysteines? Also, it was mentioned that bleaching steps were used to assess labeling, but this was not shown so it is unclear what information was being obtained from these analyses. Both of these points should be addressed to give confidence that the label is attaching in the mode described and that there are no other possibilities of non-specific labeling.

We now state in ‘ Results’, under the same labeling condition, there was little detectable fluorescent labeling in the absence of the mutant cysteine residues (Figure 1—figure supplement 1A,B). To minimize the background labelling, we removed two native cysteine residues. Removal of native cysteine residues in AdiC has been shown to have very limited impacts on its function. Furthermore, in ‘Results’ and ‘Materials and methods’, we state that the peak of the protein containing the two mutant cysteine residues, detected at 280 nm, co-migrated with that of the fluorophore dye detected at 550 nm. In contrast, no notable absorbance peak at 550 nm co-migrated with the peak of the protein without these mutant cysteine residues. Thus, there was little detectable background labeling (Figure 1-figure supplement 1A,B).

The total intensity traces, each with a bleaching step, are now shown in Figire 3—figure supplement 1 (Results).

4. Robustness of the clustering results. The clustering, as indicated in Figure 5, is not evident by eye and appears more as a continuum of conformations. It also reasons that the clustering results are dependent on the averaging, changepoint detection and k-means clustering parameters. Thus, the robustness of the identification of states should be shown across a range of parameters for each analysis step. In addition, testing different state models (e.g. 2-8 states) would be important to understand if the 4-state clustering is robustly determined or other state models also suitably fit the data.

We now state in the Results that the program examined the case of one-state distribution to the case of *k*-state distributions, one at a time in a series of separate operations. For each successful operation, the resolution criterion was ensured, i.e., peaks of all populations were separated by at least 2.5σ (‘Results’).

5. Validation of the model with mutations that are known to shift the equilibrium between states. While the results showing the linkage of the states to arginine binding corresponding to the macroscopic measurements of AdiC KD provides some supportive evidence, additional investigation that tests the different states would strengthen the proposed model. For example, can a mutation be engineered, or reagent applied that stabilizes the inward-open or outward-open state, to see that conferred in the single-molecule polarization traces?

See responses to summative comment #3.

6. A cartoon of the full gene sequence and schematic of the AdiC construct that is being used here would be useful as supplementary information, in order to understand the sizes of tags and linkers applied to the protein.

It is now provided in Figure 1—figure supplement 1C.

7. In Figure 9, the colors in the plot do not appear to match the legend (different shades of blue).

The correct color has now been used.

8. Since the data is analyzed over single trajectories, how long are these traces typically? How long do the traces have to be in order to observe enough transitions between states?

The average length of the trace is ~4 s (‘Materials and methods’), which is, on an empirical basis, adequate in terms of containing all 4 states and >15 events.

9. The fitting of the KDs for states C1 and C4 are not justified as there does not appear to be a significant change in the probabilities over the arginine titration. Therefore, they should not be reported with confidence. Along these lines, are the errors that are being reported in Table 1 on the best-fits or are they over different samples?

We now explain this issue in ‘Results’: even though the concentration dependence plot of the probability of *C*_1_ or *C*_4_ are relatively flat, their KD values are determinable. This is because the model can be fully specified by a certain combination of only even equilibrium constants, e.g., three of the six Ki,j values, three of the six  LKi,j values, and merely one of the four KD values. As such, in the present so-called over-determined case, the KD values of the relatively flat traces are fully constrained by those more curved traces in a global fit of Eq. 5 to all four plots simultaneously. The errors are confidence intervals obtained by a boot-strapping method, which is now stated in the foot note of the table.

Reviewer #2 (Recommendations for the authors):1. The key difference of the data processing described in this manuscript compared to the earlier published work is the introduction of the normalization step. Due to the random X-Y orientation of the transporter proteins on the surface, each protein molecule needs to be analyzed individually to get its own theta and phi angles in the laboratory frame (Cartesian coordinate). Normalization of the angles based on one of the identified states allows to directly compare the results for different particles directly. This is an interesting improvement, so the authors may need to mention this in the abstract.

The reported study represents the first time that a high-resolution polarization microscopy has been applied to the investigation of a membrane protein. During the struggle in the past 7 years to make this happen, we have developed methods to prepare the samples, and made a number of improvements. Three analytic advances are highlighted in the manuscript including “normalization of the angles”. Indeed, observing four states is not a mechanistic advance. However, the ability to detect them together in a membrane protein, analyze the data, and extract energetic and kinetic information about them are significant advances.

2. In principle, the normalization may improve the accuracy of the mean angle values even as the deviations might increase. It would be helpful if the authors compared the ensemble-averaged mean values and the single-trace mean values.

We now add current Figure. 9 to show the distribution of the mean angle values of individual particles.

3. From my understanding, to better differentiate the states, traces visiting all 4 states should be analyzed. When the authors select the traces with at least 12 transition events, do the events include transitions between all four states? The authors might want to clarify.

All analyzed traces are long enough and contain all four states, which is now stated in ‘Materials and methods’. The transitions among all four are observed as described in the original companion manuscript which address the kinetic aspects of the observed conformational changes. Given the present manuscript concerns primarily resolution of individual conformations and the free energy difference among them but not their kinetics, the connectivity information is not directly relevant here. We intend to share this information in a revised version of that manuscript for future publication, in which we will discuss the experimental evidence that disproves the hypothesis that AdiC is an obligatory exchanger.

4. It is unclear how the authors establish the number of distinct states to be considered in the system. Is the number of states based on the expected structurally resolved 4 states or obtained from the analysis of the traces themselves? There is a proposed so-called full occluded state. Could/should this state be included in the data analysis?

The determination of the number of states is not influenced by any preconceived model. As stated above under summative comment#2 and now in ‘Results’, four states are the highest number that we can detect while satisfying the common resolution definition for microscopy, without any preconceived kinetic model. (The "cul-de-sac” state that connect to each of the two “occluded” state in the 24-state kinetic model presented in the original companion manuscript might be interpreted as fully occluded state, which could in principle be connected by transition in a rapid equilibrium.)

5. The fluorophore is attached via two cysteines. Therefore, in the same conformation, the fluorophore has two orientations related by 180-degree rotation. Will this affect the detected signal? If not, then would a protein conformational change, which makes the fluorophore rotate by 180 degrees, be detected?

Given the dipole of the fluorophore has an inherent two-fold symmetry, it has the same spatial relation to the two cysteines. Thus, this attachment does not affect the signal. By definition, an “exact” 180-degree rotation motion around the symmetry center of the fluorophore is undetectable in any dipole-polarization-based analysis. In this rare case, a different technique is certainly needed. The present method is developed to detect small movement by resolving small angle changes. There are many techniques that can be used to detect large movement.

6. The protein is modified by a bi-functional fluorophore and surface-immobilized with multiple binding tags. How do you evaluate whether these manipulations affect the state populations and transport activity? The structures of Adic were all solved in the outward-facing states. Simulation (PLoS ONE 11(8): e0160219) also indicated that the inward-facing states, especially the inward-open state, are relatively unstable, suggesting that the population of the inward-facing states should be low. However, in Figure 7B, the inward-open state is highly populated, which seems inconsistent with the expectations. It seems that some additional experiments and or more extensive discussion are needed to support the authors' estimates of the state populations.

The first part has been responded under summative comment #3.

As now stated in ‘Results’, reconstitution of protein into lipid-containing nanodiscs, tags, attachment, mutations, and fluorophore labels are all necessary components of the present system, contributing to system errors. Currently, there are no other techniques that can assess, under the same conditions, the ultimate impact of these factors on the properties, which highlights the need to develop the present method. In terms of energetic impact, a practically relevant question would be whether the KD values estimated using the present method with all of those potentially impactful factors are comparable to those values estimated for the wild-type protein by means of an already established method, such as ITC, and the probabilities and rate constants that we estimated can predict kinetic parameters. As we outlined above, and in ‘Discission’, the Kd values that we estimated are compatible with previously reported ITC values, and in the original companion manuscript, kinetic parameters are within the expected error range of the previously reported values.

Regarding the sidedness, in the quoted study, the MD simulation of outward-facing states were based on partial AdiC structures whereas inward-facing states was based on a different transporter (GadC). The results of simulating the inward-facing with GadC appear to heavily depend on the residues introduced. In any case, Tsai et al. (Biochemistry 2012) have shown that AdiC is relatively symmetric in term of flux kinetics: comparable Km values for both sides and comparable maximal flux rates in either direction (‘Discussion’). Our model is consistent with their measurements. This point has now been discussed. We emphasize that the observed symmetric property may not occur under different experimental conditions, such as a different pH.

7. The state assignments are based on the similarity of the theta and phi angles to those calculated from the four crystal structures. To me, this does not seem entirely convincing, especially regarding the inward-facing states, expected to be lowly populated. The authors might need to use mutants or ligands to stabilize the inward-facing states to verify the assignment.

As mentioned above under summative comment #2, the open versus “occluded” states were determined on the basis of substrate dependence (‘Discussion’). Regarding the sidedness (‘Materials and methods’), the relation between the orientations determined from polarization measurements and those from the structures of known states helps distinguish between the so-called internal and external conformations. The external open and external occluded structures of AdiC are available. Fortunately, for these two states, the two methods yield consistent orientations. Furthermore, we also used the internal open and internal occluded structures of two related transporter as proxies, or space holders, for the corresponding states of AdiC in the exercises to illustrate how to perform multistate spatial alignment.

8) The authors need to describe how they estimate the errors in Table 1.

The confidence intervals were obtained by a boot-strapping method, which is now stated in the footnote of the table.

Reviewer #3 (Recommendations for the authors):1. This work monitors only a single transmembrane helix (helix 6a) in AdiC. This was presented as an advantage, but in reality it also creates strong limitations for data interpretation. By comparing the orientations of helix 6a in PM-identified states (C1-C4) and in structural states, C1-C4 were dubbed externally open (Eo), externally occluded (Ex), internally open (Io), and internally occluded (Ix) states. Such assignment comes with high uncertainty even in optimal conditions. For example, without "seeing" the rest of the protein, how do the authors know that a particular PM state is not an intermediate state not observed in structural studies? Such a concern arises in part because of the assignment of C1 as Ex and C4 as Ix, with a state possibility of 0.19 and 0.14, respectively. Why would an antiporter sample occluded states so often without ligands? Is it possible that C1/C2 are both externally open (with some differences in helix 6a orientations) and C3/C4 both internally open and the occluded states are too rare to be observed when there is no ligand?2. The assignment of PM states was actually done under not-so-optimal conditions. AdiC structures were only determined in ligand-free Eo and ligand-bound Ex states, so the authors used BasC and ApcT structures to represent the Io and Ic states of AdiC. Multiple questions about data treatments follow naturally. What's the justification to compare ligand-free PM states with ligand-bound structural states (Figure 6)? Although BasC (complexed with a nanobody) and ApcT have similar LeuT fold like AdiC, their sequences are not particularly similar and it's conceivable that AdiC's helix 6a might have significantly different orientations in Io and Ic states than the helix 6a in BasC and ApcT. How would such a possibility affect the assignment of the PM states?

The above two related comments are addressed together below.

We now discuss in the manuscript explicitly: The conformational states resolved in the present study are defined by the orientation of the dipole of a fluorophore attached to an α-helix that interacts with and is thus spatially constrained by other surrounding secondary structures (‘Results’). On the basis of structures in different states, the orientation changes of the dipole are thus expected to reflect the rotation motion of the protein.

For reference, in FRET, each of a pair of fluorophore labels is attached to a residue in a secondary structure within a domain. The transfer efficiency is being used to report the relative movement between the two domain or two proteins. Strictly speaking, each label would not directly report the behavior of the rest of the protein either. As such, one fundamental difference between FRET and PM is that FRET requires two fluorophores, and PM requires only one.

As already mentioned under summative comment#2, given that the probabilities of conformations *C*_2_ and *C*_3_ clearly varied with the Arg^+^ concentration whereas the other two states varied little, *C*_2_ and *C*_3_ should be two open states that could directly interact with substrates, and the remaining two would then likely be the occluded states (in adopting the conventional name). Regarding the sidedness of the model, the relation between the orientations determined from polarization measurements and those from the structures of known states helps distinguish between the so-called internal and external conformations. The external open and external occluded structures of AdiC are available. Fortunately, for these two states, the two methods yield consistent orientations. Thus, our data are consistent with the relation of C1 as Ex and C2 as Eo, and by symmetry, C3 as Io and C4 as Ix, which are consistent with spatial relations observed in the Io and Ix states of two transporters containing the same structural fold. If anything, we put forward a model or hypothesis for future testing, on the basis of a set of consistent relations.

To be sure, the above exercise of spatial alignment, in which internal open and internal occluded structures of two related transporter are used as proxies, or space holders, for the corresponding states of AdiC, is primarily to illustrate an algorithm for performing multi-state spatial alignment.

In the revised version of the original companion manuscript for future publication, we will discuss the experimental evidence that disproves the hypothesis that AdiC is an inherently obligatory substrate-exchanger in a transparent, analytic manner.

3. Another main issue is about the relation of ligand-free and ligand-bound PM states. There are four ligand-bound PM states, labeled as C1L, C2L, C3L, and C4L to indicate that they correspond to the ligand-free C1, C2, C3, C4 states. However, the ligand-free and ligan-bound PM states appear to have quite different θ/ψ profiles (Figure 5), and the latter were not compared to the structures. How did the authors link ligand-bound PM states to the structural states? How to know for sure that C1 is related to C1L, C2 to C2L, etc? Moreover, it could be misleading to distinguish the corresponding ligand-bound and -free states just by an "L" as if their only difference is the ligand – the technique here provides no such resolution.

We now show in Figure. 5 the probability density of each state plotted against *θ* and *φ*, and in Figure. 6 in the mean angle values, for many Arg^+^ concentrations in the tested range. The probability densities change but the angle values do not exhibit systematic variations across the concentration range. In a model based on the PM data, the states are then denoted as C_i_ and C_i_L as traditionally done. This type of elementary notations in an energetic model is consistent with the Monod-Wyman-Changeux model (J. Mol. Biol., 1965) for ligand-dependent conformational mechanism of allosteric proteins in that all conformations of a protein occur in the absence of ligand, and the binding of a ligand does not create a new conformation but merely energetically stabilizes a spontaneous conformation that is ready to capture the ligand (‘Results’).

4. The authors might want to consider utilizing additional strategies to verify their state assignments. One possibility is to use structures as guidance to lock AdiC in a particular state via Lysine crosslinking and see how such maneuver affects PM results. The determined K_D_ reflects how ligands drive the protein from ligand-free to ligand-bound conformational equilibria without providing much insight into the conformational details in each set of equilibrium

See responses to summative commnet#3.

Reviewer #4 (Recommendations for the authors):1) On page 20, the authors note that they set a threshold to filter out molecules whose total intensity varies during the measurements. The statement that "while fluorescence intensity is expected to vary among different polarization directions, the total intensity should be essentially invariant" is not true. Since the authors use TIRF illumination to excite the molecules, the excitation polarization component along the tilting direction (e.g., along the y-axis) of the excitation is 0, i.e., molecules oriented along that direction (e.g., y-oriented) will be excited less effectively compared to other orientations.

We originally intended to mean implicitly a particle with large total intensity changes due to such a factor as fluorescence bleaching in more than one step would be excluded, because this phenomenon would most probably reflect the attachment of more than one fluorophore to a single protein molecule. We have now stated this point clearly.

2) Could the authors provide more details regarding how the clusters are ranked? The authors note that C1-C4 are "ranked according to the values of both angles". It is not clear to me how this is done. Also, what is the range of the measured theta_L and phi_L? And how is the warping of the spherical coordinates handled in the ranking process, e.g., a change from 350 deg to 10 deg is +20 deg or -340 deg.

If stated literally, the conformational states are numerated according to the sequence in the solution to achieve the shortest pathway among the four states evaluated in a Cartesian system, denoted as *C*_1_, *C*_2_, *C*_3_ and *C*_4_. This operation was done after assigning the angle values. We now make this clear in ‘Results’, and remove the use of angles as proxies.

*θ* is defined from 0° to 90° and *φ* from 0° to 180° with degenerate solutions of *θ*, 180° – *θ* and *φ*, 180° + *φ*, which was given in a previous paper, and is now included in ‘Materials and methods’. Because the angle changes predicted from the structures are small, about 10 – 40°, a change of +20º change instead of -340º would be chosen.

3) Is the k-means clustering also based on the distance in the Cartesian space, similar to the state identification?

Yes. We now explicitly explained this in ‘Results’ and in ‘Material and methods’.

Following comment (1) from above, could the authors comment on the possibility of further improving the measurement precision and accuracy using the excitation-dependent total intensity? Since the authors report a wobble angle of 22.5 degrees, the excitation dipole moments should be mostly aligned with the emission dipole moments.

As the reviewer implied, we could use multiple linear excitations of different polarizations in a multiplexing operation, with 8 to 16 channels. Analyzing the intensity changes caused by different excitations at different emission channels could in principle improve somewhat the precision and accuracy of *θ* determination near 0º and 90º. We chose to use a circularly polarized laser because it is technically much easier to implement, and has thus far been adequate for our purposes. Importantly, a simpler system will also be easier for other labs that are interested in this technique to adopt. Note that the use of circular polarization eliminates the confounding slow wobble of the fluorophore that leads to a difference in the average orientations of the excitation and emission dipoles during individual sampling periods.